# Pharmacological Treatments for Cocaine Craving: What Is the Way Forward? A Systematic Review

**DOI:** 10.3390/brainsci12111546

**Published:** 2022-11-14

**Authors:** Dângela Layne Silva Lassi, André Malbergier, André Brooking Negrão, Lígia Florio, João P. De Aquino, João Maurício Castaldelli-Maia

**Affiliations:** 1Interdisciplinary Group of Alcohol and Drug Studies (GREA), Department and Institute of Psychiatry, Medical School, São Paulo University, São Paulo 05403-010, SP, Brazil; 2Department of Psychiatry, Yale University School of Medicine, New Haven, CT 06510, USA; 3Department of Neuroscience, Medical School, ABC Health University Center, Santo André 09060-870, SP, Brazil; 4Department of Epidemiology, Mailman School of Public Health, Columbia University, New York, NY 10032, USA

**Keywords:** craving, cocaine, pharmacological treatment, review, target

## Abstract

Background: cocaine craving is a core feature of cocaine use disorder and remains a critical challenge for abstinence and relapse prevention. This review summarizes the anti-craving efficacy of pharmacotherapies tested for cocaine use disorder, in the context of randomized-controlled clinical trials. Objectives: we assessed the databases of the U.S. National Library of Medicine, Google Scholar, and PsycINFO, without date restrictions up to August 2022, to identify relevant studies. Study eligibility criteria, participants, and interventions: we included double-blinded randomized-controlled trials investigating pharmacotherapies for cocaine craving and/or cocaine use disorder whose outcomes included cocaine craving. Study appraisal and synthesis methods: Two authors screened studies’ titles and abstracts for inclusion, and both read all the included studies. We systematically gathered information on the following aspects of each study: title; author(s); year of publication; sample size; mean age; sample characteristics; study set-ting; whether participants were treatment-seeking; study design; craving measures; study interventions; drop-out rates; and other relevant outcomes. Results: Overall, we appraised 130 clinical trials, including 8137 participants. We further considered the drugs from the studies that scored equal to or greater than six points in the quality assessment. There was a correlation between craving and cocaine use outcomes (self-reports, timeline follow-back or urinary benzoylecgonine) in the vast majority of studies. In the short-term treatment, acute phenylalanine-tyrosine depletion, clonidine, fenfluramine, meta-chlorophenylpiperazine (m-CPP) and mecamylamine presented promising effects. In the long term, amphetamine, biperiden, carbamazepine, lisdexamfetamine, lorcaserin, methamphetamine, mirtazapine, pioglitazone, progesterone, guanfacine, levodopa, nefazodone presented promising anti-craving effects. Unfortunately, the highly tested medications were not successful in most of the trials, as follows: propranolol in the short term; amantadine, aripiprazole, bromocriptine, citicoline, ketamine, modafinil, olanzapine, topiramate in the long term. The remaining 52 medications had no positive anti-craving outcomes. Limitations: Our review was limited by high heterogeneity of craving assessments across the studies and by a great range of pharmacotherapies. Further, the majority of the studies considered abstinence and retention in treatment as the main outcomes, whereas craving was a secondary outcome and some of the studies evaluated patients with cocaine use disorder with comorbidities such as opioid or alcohol use disorder, schizophrenia, bipolar disorder or attention deficit hyperactivity. Lastly, most of the studies also included non-pharmacological treatments, such as counseling or psychotherapy. Conclusions: There is a direct association between craving and cocaine use, underscoring craving as an important treatment target for promoting abstinence among persons with cocaine use disorder. Clonidine, fenfluramine and m-CPP showed to be promising medications for cocaine craving in the short-term treatment, and amphetamine, biperiden, carbamazepine, lisdexamfetamine, lorcaserin, methamphetamine, mirtazapine, pioglitazone, progesterone, guanfacine, levodopa, nefazodone in the long-term treatment.

## 1. Introduction

Cocaine use disorder causes a host of medical, psychological, and social problems worldwide, including cardiovascular disease, infection, violence, and crime. The United Nations Office on Drugs and Crime [1] estimates that, in 2018, 19 million people used cocaine, a number that is expected to grow against the backdrop of the socioeconomic crisis caused by SARS-CoV-2 pandemic. Yet, despite decades of clinical research, thus far no pharmacological treatments for cocaine use disorder have been established. One of the most critical barriers to abstinence and relapse prevention is craving [2], a core symptom of cocaine use disorder, and one whose clinical relevance is underscored by its inclusion as a diagnostic criterion to diagnose stimulant use disorders in the 5th Edition of the Diagnostic and Statistical Manual of Mental Disorders (DSM-5) [3].

At the neurobiological level, prolonged cocaine use can cause brain circuitry modifications and progressively sensitize dopamine systems, leading to recurrent and intense urges—or craving—to use cocaine. Craving is defined as “a more intense, urgent abnormal desire characterized by longing, yearning and physiological need for drug” [4]. The shift from “wanting” to use a drug to “craving” it occurs due to a progressive salience attributed to drug-related stimuli, even when negative cognitions are attributed to drug use itself [5]. Craving can be elicited by cocaine cues—such as paraphernalia and places/situations related to cocaine use—by acute withdrawal, by cocaine, and by stress, which makes feasible to evaluate in a naturalistic way with an ecological momentary assessment or in laboratory studies with the induction of craving. 

Distinct biological and psychological mechanisms are believed to contribute to various types of craving [6]. Some evidence indicates that specific types of craving may be particularly amenable to pharmacological interventions, such as the craving for compulsive use in early abstinence [7].

Although the precise pathophysiology of craving remains elusive, some of its neural substrates have been identified, including progressive alterations in limbic, striatal and cortical systems. The amygdala plays a critical role in learning the relationship between significant stimuli and the signals that anticipate them; the anterior cingulate connects to the amygdala and subserves mood regulation and emotional responsivity; the nucleus accumbens mediates the reinforcing properties of cocaine [6]; and the basal ganglia are believed to underlie compulsive component of addiction. Altogether, the shift of cognitive and emotional aspects of cocaine-related memory can lead to compulsive drug-seeking behavior [8].

Converging functional neuroimaging studies involving persons who use cocaine show cue-elicited increases activity in limbic regions, such as the amygdala and anterior cingulate, cerebellum, and prefrontal cortex, as well as decreased activity in subcortical regions, such as the basal ganglia [6,8,9,10]. Another study with positron emission tomography (PET) [9] using personalized cues and autobiographical memories found a decreased activity in the prefrontal cortex, lending support to previous findings showing a disruption of prefrontal activity during craving [4].

Although multiple clinical trials have investigated pharmacological interventions for cocaine use disorder, thus far few of them have included craving as a treatment efficacy outcome. Given the clinical relevance of craving as a potential mediator of return to using cocaine, there is growing interest in studying stimulants, antidepressants, and anticonvulsants as anti-craving pharmacotherapies, as reviewed in recent metanalysis [11,12,13,14]. 

As craving is strongly associated with compulsive cocaine use [15,16] and is a significant predictor of relapse [2], it is an important treatment target for persons with cocaine use disorder. Hence, we have summarized and appraised findings from randomized trials investigating anti-craving medications for this clinical population. To our knowledge, this is the first systematic review to specifically synthesize and appraise the evidence for anti-craving effects of pharmacological interventions for cocaine use disorder.

## 2. Methods

### 2.1. Eligibility

Original pharmacological studies including persons with cocaine abuse or dependence were included in this review if craving was a treatment outcome. Non-original reports such as reviews, meta-analyses, case reports, discussion articles, study protocols; studies that were not randomized, controlled and double-blinded; studies in languages other than English; preclinical studies; studies whose samples was not composed by persons who use cocaine; and studies that did not include craving as a treatment outcome were excluded. Given the variety of nosological classification of cocaine use disorders across the studies included in our review, we use the terms cocaine abuse and dependence interchangeably.

### 2.2. Information Sources 

We searched the databases of the US National Library of Medicine, Google Scholar and PsycINFO up to 1 August 2022 to identify relevant studies.

### 2.3. Search Strategy

The keywords used in the search were ‘craving’ and ‘cocaine’ and ‘random* controlled Blind*’. The filters used in PubMed and PsycINFO, were ‘clinical trial’ and ‘humans’. The detailed search strategy is included in the Appendix A.

### 2.4. Study Selection

First, two authors (D.L.S.L. and J.M.C.M.) read the abstracts of all studies found in the initial search (*n* = 466). Duplicates (*n* = 149) and studies that were unrelated to cocaine (*n* = 77) were excluded. Second, these authors read the remaining studies independently (*n* = 240). After eligibility were applied, 110 articles were excluded. The remaining 130 studies were included in this review. The present review followed the Preferred Reporting Items for Systematic Reviews and Meta-Analyses (PRISMA) Guidelines Checklist [17] for transparent reporting of systematic reviews and meta-analyses, as presented in Figure 1.

### 2.5. Assessment of Methodological Quality

Two authors, D.L.S.L. and J.M.C.M. independently assessed the risks of bias in each study included, using the Study Quality Guide from the Cochrane Consumers and Communication Review Group [18].

### 2.6. Data Collection Process

Two authors, D.L.S.L. and J.M.C.M., read all the 130 included studies independently. D.L.S.L. tabulated the data, which J.M.C.M. evaluated. Disagreements between the two authors were resolved by discussion and consensus. 

### 2.7. Data Items

We systematically gathered information on the following aspects of each study: title; author(s); year of publication; sample size; mean age; sample characteristics; study setting; whether participants were treatment-seeking; study design; craving measures; study interventions; drop-out rates; and other relevant outcomes.

## 3. Results

### 3.1. Distribution

The included studies ranged from 1987 to 2022. A total of 92% of the studies (120) were conducted in the USA; 2,4% (3) in Brazil; 2,4% (3) in the Netherlands; 0,8% in each of the following Australia (1), Canada(1), France (1) and Mexico (1).

### 3.2. Participants

Combined, the studies included had 8137 participants; the smallest study had eight participants, whereas the largest had 358 participants. Men comprised the majority (75%) of the sample and 21 studies had only male participants. Conversely, only one study enrolled exclusively women. Seven studies did not report the participant’s gender. The median age was 39 years.

While in 50% of the studies, participants were treatment-seeking, 40% of the studies did not describe participants’ intention to engage in treatment. Approximately 27% of studies included participants who used cocaine through the smoked route; 2% of studies participants who used cocaine intravenously; and 58% of studies included participants who used cocaine through more than one route of administration. Information on the route of administration on the remaining 12% of studies was lacking.

### 3.3. Assessments 

Ninety-three percent of the included studies used the Diagnostic and Statistical Manual of Mental Disorders criteria for cocaine abuse or dependence (20% DSM-III, 71% DSM-IV, 2% DSM-5). The other 7% included criteria from International Classification of Diseases (ICD-10) and unstructured self-report measures.

Craving was assessed by single or multi-factorial craving scales. Approximately 51% of the studies used a Visual Analog Scale (VAS), either alone or in combination with another semi-structured assessments. The most frequent semi-structured assessments were: The Addiction Severity Index (ASI) (39%); the Cocaine Craving Questionnaire (CCQ) (20%); the Brief Substance Craving Scale (BSCS) (20%); the Minnesota Cocaine Craving Scale (MCCS) (16%); and the Cocaine Selective Severity Assessment (CSSA) (12%). These measures are summarized in Appendix A, in the Appendix A.

### 3.4. Quality Assessment

The quality assessment was performed using the Study Quality Guide from the Cochrane Consumers and Communication Review Group [18], considering six domains of risk of bias: random sequence generation; allocation concealment; blinding of participants and personnel; blinding of outcome assessment; incomplete data outcome; and selective reporting. 

We used a previously established scoring system, in which low risk is indicated by 2 points, unclear risk 1 point, and high risk 0 points, as shown in Appendix A, in the Appendix A. The scores of the studies included ranged from 2 to 12 points: 85% (111 studies) scored 6 or more (only one scored 12 points, and only one scored 2 points). 

### 3.5. Interventions and Outcomes

Setting: 74%, 21% and 5% of the studies were conducted in the outpatient, inpatient, and hybrid settings, respectively.

Duration of intervention: The duration of the intervention ranged from single-dose/one-day intervention to 16 weeks. The most prevalent duration of intervention was 12 weeks (21% of studies). Most (92%) studies compared a pharmacological agent to placebo, and 8% of studies compared two or more active drugs, directly and indirectly.

Adjuvant interventions: Over 60% of trials (74) offered—in addition to the study intervention—treatment as usual, such as individual counseling, cognitive behavioral counseling, or 12-step based approaches.

Efficacy: The majority of trials—46% or 60 studies—found no significant improvement in craving measures by time or group intervention. In approximately 21% of trials, or 27 studies, the active drug intervention suppressed craving and differentiated from placebo. 

A total of 84 studies evaluated both craving and cocaine use as outcomes, 76% of them (64) presented a direct correlation between craving and cocaine use after the treatment. The other studies (20) showed divergent results.

First, we separated the studies into two large groups according to the duration of intervention: Acute interventions when up to 7 days and Sub-Acute interventions for more than 7 days. Within each large group, pharmacotherapies were divided into pharmacological classes: antidepressants, antipsychotics, psychostimulants, and other drugs. Detailed data is depicted in Table 1, Table 2, Table 3, Table 4 and Table 5.

### 3.6. Acute Interventions

#### 3.6.1. Antidepressants

One study administering acute fenfluramine [22] found evidence of anti-craving effects 2-fold greater than placebo. 

#### 3.6.2. Psychostimulants

Acute administration of serotonergic (5HT) agonist meta-chlorophenylpiperazine (m-CCP) [39] led to a substantial reduction (20%) in craving. The 5HT agonist lorcaserin in a single dose showed mixed results; it had anti-craving effect only after placebo IV doses, but not after IV cocaine doses [58].

#### 3.6.3. Other Drugs

The dopaminergic (DA) precursor depletion method (APTD) significantly reduced both cue- and cocaine-induced craving [112]. Two trials of noradrenergic (NA) agonists—one with clonidine [103] and one with guanfacine [99]—resulted in improvement in craving compared to placebo, while another study with guanfacine reported negative results [121]. Finally, the beta NA blocker propranolol, compared to placebo, significantly reduced craving in a single-dose study [134].

### 3.7. Sub-Acute Interventions

#### 3.7.1. Antidepressants

No study among the six trials involving selective serotonin reuptake inhibitors (fluoxetine, paroxetine, fenfluramine) found evidence of anti-craving effects [21,25,27,28,33,36]). Among the trials involving dual action in 5HT and NA signaling (venlafaxine, citicoline, nefazodone, desipramine) [20,23,24,26,29,30,31,34], two found a statistically significant reduction in cocaine craving: one study administering long-term desipramine [26], and one study administering long-term nefazodone [24]. Mirtazapine 30 mg daily showed significant anti-craving effect in one study [32], while no effect in craving in another trial [19]. Lastly, bupropion [35] did not have demonstrable craving suppression effects. 

#### 3.7.2. Antipsychotics

Aripiprazole led to a significant reduction in craving in one study [66], while another two studies [70,71] did not differentiate from placebo. Studies administering olanzapine [65,67,68,72], quetiapine [76] and risperidone [65,69,74] did not find evidence of craving suppression effects either. When compared to haloperidol, olanzapine demonstrated significantly better results in one trial [75], while another showed that haloperidol [73] was significantly better in reducing craving than olanzapine. 

#### 3.7.3. Psychostimulants

Among studies administering modafinil, two [38,51] out of seven studies [38,42,43,44,46,51,52] found significant improvements in cocaine craving; as did another three studies administering amphetamine [48], lisdexamphetamine [55] and methamphetamine [54]. Two trials testing the use of lorcaserin 20 mg daily [49,60] showed significant decreases in craving compared to placebo. Studies with atomoxetine [40,64], dexamphetamine [56,62], diethylpropion [37], mazindol [57,63,127], methylphenidate [47,53,59,61], ritanserin [41,45,50] did not demonstrate anti-craving effects.

#### 3.7.4. Anticonvulsants

Only one study [80] of two [23,80] comparing carbamazepine to placebo showed a significant decrease in craving. The anticonvulsant topiramate was found to have anti-craving effects in two [81,82] out of four trials [81,82,83,86]. The trials with baclofen [141], gabapentin [77], lamotrigine [78], phenytoin [79], tiagabine [87], valproate [72,84] and vigabatrin [144] reported no significant effect on cocaine craving in comparison with placebo. 

#### 3.7.5. Other Drugs

Nicotinergic agonists (nicotine and varenicline) were investigated in two trials, and an antagonist (mecamylamine) in two studies: mecamylamine significantly reduced craving in one trial [85], while in another [131] only the placebo group showed significant reduction; transdermal nicotine significantly increased cue-induced craving in comparison to placebo in one trial [145] while varenicline [126] did not differ from the placebo. Biperiden, a muscarinic antagonist, reduced craving by 37.6%, compared to a 19% change from baseline produced by the placebo [96]. Bromocriptine led to a significant reduction in craving in one study [92], while another two [97,102] did not differentiate from the placebo. The DA precursors levodopa/carbidopa, administered for more than 5 weeks and combined with non-pharmacological therapies, were found to reduce craving in one [138] of two studies [120,138], in which it was administered for a longer period (12 weeks) and combined with cognitive behavioral therapy. The administration of amantadine was found to have anti-craving effects in one [140] of three [101,109,140] studies; as citicoline did in one [132] out of three studies [90,113,132]. Two [100,117] studies in three [100,117,146] comparing 7 days of allopregnanolone to the placebo showed a significant decrease in craving. Ketamine showed a significant craving reduction in one [93] of the three studies [93,94,95]. The PPAR (Peroxisome proliferator-activated receptor)-Gamma agonist Pioglitazone reduced cocaine craving by a factor of b = 0.24 when compared to a factor of b = 0.09 for participants receiving placebo [136].

Oral naltrexone, an opioid receptor antagonist, did not show any difference in craving in comparison to the placebo [118,137]; as well as the glutamatergic partial agonist D-cycloserine [106,128,129,135]. Propranolol, compared to the placebo, did not reduce craving in long-term studies [104,108,134]. The Calcium Channel Antagonists amlodipine [115], isradipine [105] and nimodipine [133]; acamprosate [147]; the Cocaine esterase RBP- 8000 [123]; the Cox-2 inhibitor celecoxib [130]; disulfiram [125]; ecopipam [122]; L-tryptophan [91]; lidocaine [88]; memantine [89]; the corticosteroids metyrapone [107,110]; N-acetylcysteine [139,143,148]; pergolide [114]; pramipexole [124]; reserpine [142]; riluzole [25], selegiline [98] and cannabidiol [116,119] showed no differences compared to the placebo either. In a study, participants who received oxytocin experienced more severe cocaine craving relative to the placebo [111].

### 3.8. Treatment Dropout

Attrition varied widely among clinical trials, ranging from 0 to 82%, with an average attrition of 40%. Twenty-seven trials did not report their dropout rate. Among the studies with outpatients, 32% showed more than half of the sample dropped out at the end of intervention, against 14% of the inpatient trials. Half of the studies with 5HT agents had an attrition rate greater than 50%. Notably, studies that included adjuvant non-pharmacological therapies had lower attrition rates.

## 4. Discussion

We sought to systematically review and appraise the evidence base on pharmacological interventions for cocaine craving, an important treatment target for cocaine use disorder. Altogether, we appraised 130 clinical trials examining a wide range of compounds, with several mechanisms of action. From the analysis of the current review, we observed correlation between craving and cocaine use outcomes (self-reports, timeline follow-back or urinary BE) in the vast majority of studies. Therefore, we can consider craving as an intermediate target for achieving abstinence.

We highlighted here the promising drugs presented in the studies that scored equal to or greater than six points in the quality assessment. In the short-term treatment, acute phenylalanine-tyrosine depletion (1 study with significant positive anti-craving effects in 1 study included in the present review), clonidine (1/1), fenfluramine (1/1), and m-CPP (1/1), mecamylamine (1/2) presented promising effects. In the long term, lorcaserin (2/2) amphetamine (1/1), biperiden (1/1), carbamazepine (1/1), lisdexamfetamine (1/1), methamphetamine (1/1), pioglitazone (1/1), progesterone (2/3), guanfacine (1/2), levodopa (1/2), mirtazapine (1/2) and nefazodone (1/2) presented promising anti-craving effects. Unfortunately, the highly tested medications were not successful in most of the trials, as follows: propranolol (1/3) in the short term; amantadine (1/3), aripiprazole (1/3), bromocriptine (1/3), citicoline (1/3), ketamine (1/3), modafinil (2/7), olanzapine (1/4), topiramate (1/4) in the long term.

Craving can be triggered by the drug itself, by cues related to drug use, by stress, and by withdrawal. It may occur in the period of immediate abstinence or in a longer period of abstinence. In the first process, there is an immediate activation of limbic system related to an anticipatory state and reward expectancy, and a disruption of medial prefrontal activity leading to compulsive drug-related behavior [4]. Despite having a low number of studies, clonidine, fenfluramine and m-CPP showed to be promising medications for acute anti-craving effects, based on the experimental studies (i.e., cue-induced, stress-induced) included in this review.

The craving that occurs after a longer period of abstinence, on the other hand, can be better explained by the incubation mechanism, in which even after drug use suspension and exposure to drug cues, there is a progressive increase in craving, remaining high within 3 months of abstinence, with a reduction after 6 months and is related to neuroadaptations and increased brain-derived neurotrophic factor (BDNF) levels [149]. The most promising drugs for such chronic purposes came from the longer randomized controlled trials included in this review. The class of psychostimulants had several options such as amphetamine, lisdexamfetamine, lorcaserin and methamphetamine for chronic cocaine craving. Our findings are in line with a review [150] of 9 clinical trials, of which 4 evaluated craving and one showed a significant anti- craving effect of dexamphetamine. Progesterone was shown to be effective for treating craving in two out of three studies, in cocaine-dependent men and women who used the drug for 7 consecutive days. The antidepressant mirtazapine also showed a significant reduction in craving in 1 of 2 studies with it. Carbamazepine, biperiden and pioglitazone also presented significant anti-craving effects in long term treatments, although each one was evaluated once.

There are currently no approved drugs for the treatment of cocaine use disorder, with psychosocial intervention still being the standard approach. Recommendations of United Nations Office on Drugs and Crime [151] state that stimulant medications, such as amphetamines, may have modest effects on withdrawal and craving suppression for patients with stimulant use disorder. These recommendations refer to stimulants as “replacement” or “substitution therapy” and highlight studies that found evidence of craving suppression following the administration of extended-release formulations of methamphetamine 30 mg/day and lisdexamphetamine 70 mg/day. Findings of this review endorse these recommendations, but not the recommendations of The European Monitoring Centre for Drugs and Drugs Addiction, which stated in a summary of systematic reviews [152] that antipsychotics are the most prominent anti-craving drugs, although this could not be established as effective treatment. Several studies have been conducted to evaluate the efficacy of pharmacotherapies regarding dual substance disorders: some reports indicated that buprenorphine decreases cocaine use [153,154,155] in opioid-dependent humans who were concurrently abusing cocaine; one in particular [155] evaluated craving and demonstrated a significant reduction in its rate. Recent studies examined the therapeutic potential of orexin receptor antagonists in rodent models of cocaine use disorder, in reducing cocaine seeking behavior [156,157], pointing to future directions in studies in clinical settings.

We found that most studies made direct comparisons of one individual compound to placebo, or indirect comparisons between more than one individual compound and placebo. None of the studies investigated the anti-craving effects of combined pharmacological interventions. This is in stark contrast with the complex, multi-system pathophysiology of craving, which may require a combination of pharmacotherapies with complementary mechanisms [2,4,5]. This is exemplified by a randomized, placebo-controlled trial investigating the combination of topiramate and sustained release amphetamine among 127 persons with cocaine dependence, which found significant anti-craving effects over 12 weeks [158]. A pre-clinical study [159] suggested that the combination of buprenorphine and naltrexone decreases compulsive cocaine self-administration with minimal liability to produce opioid dependence and may be useful as a treatment for cocaine addiction. Hence, multi-function therapies maybe represent a more promising avenue of intervention.

Moreover, as craving is elicited by cues, stress, drug and withdrawal, studies must include, besides laboratory assessment, Ecological Momentary Assessment (EMA) due to the temporal association between exposure to drug cues, or subjective experiences, allowing for a more dynamic investigation of therapeutic effects of anti-craving treatments.

### Limitations

To our knowledge, this is the first systematic review of pharmacotherapies targeting cocaine craving. Our review has notable strengths. First, published and widely accepted guidelines to conduct and report systematic reviews were used [17]. A highly sensitive search strategy was used across several electronic databases, which yielded multiple studies providing data on anti-craving efficacy of pharmacotherapies. Further, 2 independent reviewers performed all stages of the review, with good interrater reliability.

Previous reviews [11,12,13,14] reported challenges in systematically assessing anti-craving effects across multiple studies, primarily due to a high heterogeneity of craving measures. For instance, some studies used standardized scales whereas other studies used Visual Analogue Scales (VAS), or non-standardized questionnaires, some with unclear psychometrical properties. A study [16] comparing unidimensional and multidimensional craving scales suggested that the latter had higher predictive validity for relapse of cocaine use and for treatment dropout. Further, a review of craving measures [7] even suggests that researchers use different scales within the same study due to its complexity. Still, our review was limited by high heterogeneity of craving assessments across studies and by a great range of pharmacotherapies. Further, the majority of the studies considered abstinence retention as the main outcome in cocaine treatment, whereas craving was a secondary outcome; also, some of the studies evaluated patients with cocaine use disorder with comorbidities such as opioid or alcohol use disorder, schizophrenia, bipolar disorder or attention deficit hyperactivity Lastly, most of the studies also included non-pharmacological treatments, such as counseling or psychotherapy. Nonetheless, the results of the review help bring clarity to the mixed findings on anti-craving pharmacotherapies reported in multiple clinical trials.

## 5. Conclusions

There is a direct association between craving and cocaine use, underscoring craving as an important treatment target for promoting abstinence among persons with cocaine use disorder. Given its clinical relevance, future studies aiming to develop pharmacotherapies for cocaine use disorder must consider craving as a therapeutic outcome, employing multimodal and standardized assessments—neurophysiological biomarkers, cue reactivity, ecological momentary assessment—to study this complex experience.

Despite most of the studies evaluate craving as a secondary outcome and having a low number of studies per medication in most of the cases, clonidine, fenfluramine and m-CPP showed to be promising medications for cocaine craving in the short-term treatment. In addition, amphetamine, biperiden, carbamazepine, levodopa, lisdexamfetamine, lorcaserin, guanfacine, methamphetamine, mirtazapine, pioglitazone, progesterone, nefazodone exhibited anti-craving effects in long-term treatment. Future trials targeting craving as the main outcome should include these medications trying to replicate their preliminary positive results.

## Figures and Tables

**Figure 1 brainsci-12-01546-f001:**
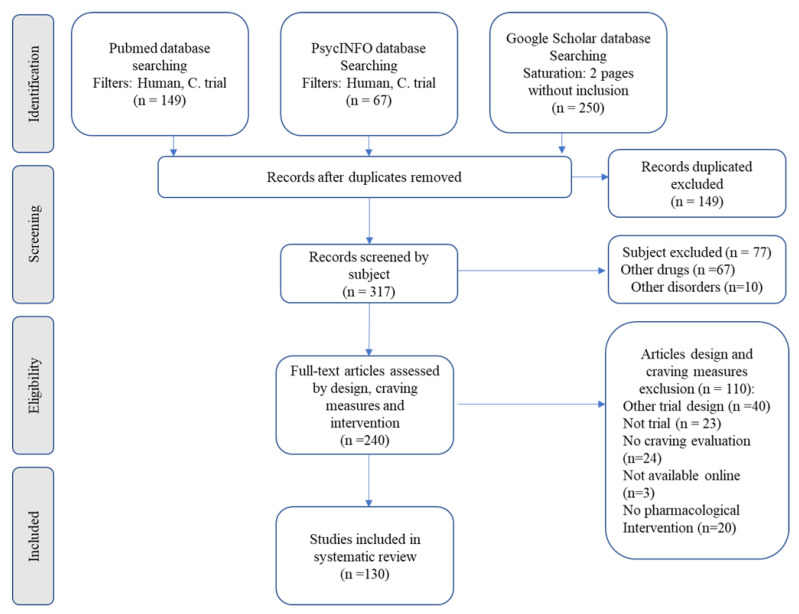
Preferred Reporting Items for Systematic Reviews and Meta-Analyses (PRISMA) Flow Diagram.

**Table 1 brainsci-12-01546-t001:** Characteristics and main findings of the studies included in the systematic review–Antidepressants.

Author, Year, Country	N (Sample); Age (Mean); Gender (M:F); Search for Treatment (Y: Yes; N: No); Setting (O: Outpatient; I: Inpatient; M: Mixed); Follow-up	Dependence; Route of Cocaine Use (S: Smoked, I: Inhaled, IV: Intravenous, O: Oral); Cocaine Use Status (C: Current Use; A: Abstinent)	Craving Assessment Instrument	Pharmacotherapy and Duration; Non-Pharmacological Intervention	Outcomes (AG: Active Group; PG: Control/Placebo Group)	Dropout %(Number); Intention to Treat (ITT) Analysis (Y: Yes; N: No)
Afshar et al., 2012, USA [19]	24; 45; 17:7; NA; O; 12 weeks	Cocaine dependence and major depression (DSM-IV); S, I; C	Cocaine Craving Scale (CCS); Cocaine Craving Questionnaire (CCQ)	Oral mirtazapine (MTZ) 45 mg or placebo (PBO) daily for 12 weeks; weekly hour-long sessions involving (manual guided) relapse prevention counseling	CCS: The time effect for the strength of craving score was significant for the placebo group [F(10,91.2) = 4.6, *p* < 0.0001], but not for the mirtazapine group (Baseline: PBO: 52.6 (29.8), MTZ: 51.1 (22.8);endpoint: PBO: 30.4 (27.3), MTZ: 45.0 (15.5)); CCQ: neither the treatment effect nor the treatment × time interaction was significant, and the within-group time effects were also insignificant (Baseline: PBO: 168.6 (56.9), MTZ: 184.6 (96.0); endpoint: PBO: 140 (57.4), MTZ: 156.4 (35.2))	NA; NA
Arndt et al., 1992, USA [20]	79; 41; 79:0; NA; O; 12 weeks of intervention + contacts after 6 months	Cocaine dependence (DSM-III) among methadone-maintained patients; S, I, IV; C	Cocaine Craving Scale (CCS)	Oral desipramine (DSP) 250–300 mg or placebo (PBO) daily and regular methadone treatment for 12 weeks; NA	CCS: Cocaine craving dropped significantly from 13 to 8 (on a scale of 1 to 20), but there was no significant difference between groups (Baseline: PBO: 12, DSP: 13; endpoint: PBO: 7, DSP: 8)	25% (20); NA
Batki et al., 1996, USA [21]	32; 34; 21:11; Y; O; 12 weeks	Cocaine dependence (DSM-IIIR); S, IV; C	Quantitative Cocaine Inventory (QCI)	Oral fluoxetine 40 mg or placebo daily for 12-week; NA	QCI: No differences in craving were found between the two groups over weeks	59% (19); NA
Buydens-Branchey et al., 1998, USA [22]	19; NA; 19:0; Y; I; 1 day	Cocaine dependence (DSM-IV); NA; A	VAS (100-mm line where 0 indicated that the chances were nil and 100 indicated that they were certain they would use cocaine)	Single dose of oral fenfluramine 60 mg or placebo; NA	VAS: a significantly greater craving decline after fenfluramine administration (*p* <0.05) compared to placebo	NA; NA
Campbell et al., 2003, USA [23]	146; 33; 102: 44; NA; O; 8 weeks	Cocaine abuse (Psychiatric Diagnostic Interview-Revised- PDI-R); NA; NA	Halikas-Crosby Drug Impairment Rating Scale (Hal-DIRS)	Oral desipramine 200 mg, carbamazepine 800 mg or placebo daily for 8 weeks; therapy	Hal-DIRS: subjects improved over time, but there was no statistically significant difference between groups	65% (95); Y;
Ciraulo, Knapp et al., 2005, USA [24]	69; 40; 49:20; NA; O; 8 weeks	cocaine dependence and had Hamilton Depression Scores of 12 or higher (DSM-IV); S, I, IV; C	Cocaine Craving Scale (CCS)	Oral nefazodone 400 mg daily or placebo for 8 weeks; individual counseling	CCS: significant reduction in craving strength for both groups (<0.0001), but more for nefazodone (*p* = 0.049). Decrease in intensity was significant in both nefazodone (*p* < 0.0001) and placebo (*p* < 0.0001) groups	26% (18)
Ciraulo, Sarid-Segal et al., 2005, USA [25]	S1: 65; 40; 47:18; NA; O; 8 weekS2: 60; 43; 43:17; NA; O; 8 weeks	Cocaine dependence (DSM-IV); NA; C	S1 and S2: Brief Substance Craving Scale (BSCS)	S1: oral paroxetine 20 mg daily or pentoxifylline 400 mg three times a day or riluzole 50 mg twice daily or placebo twice daily for 8 weeks; cognitive behavioral counseling; S2: oral venlafaxine (50 mg three times a day) or pramipexole (0.5 mg three times a day) or placebo, three times daily for 8 weeks; cognitive behavioral counseling	S1 and S2: BSCS: Changes between baseline and end-point values were not found to differ among the placebo and the active medication groups, but significant within-group decreases in mean scores in all groups	S1: 11% (7); S2: 36.6% (22); NA
Gawin et al., 1989, USA [26]	72; 29; 55:17; Y; O; 6 weeks	Cocaine dependence (DSM-III-R); S, I, IV; C	Cocaine Use Inventory and Craving Scale (−10 represents no craving; 0, about the usual craving; and 10, more craving than ever)	Oral desipramine hydrochloride 2,5 mg/Kg or lithium carbonate 600 mg or placebo daily for 6 weeks; weekly individual psychotherapy	Craving: statistically significant differences in craving appearing in the desipramine-treated subjects by week 4 (*p* < 0.0001)	54% (39); NA
Grabowski et al., 1995, USA [27]	S1: 155; 32; 112:43; 2-week runup+ 12-week intervention; S2: 21; 39; 16:5; NA; O; 2-week stabilization + 8 weeks of intervention	S1: cocaine-dependent (DSM-III); S2: cocaine and opiate dependent in methadone maintenance treatment (DSM-III-R); NA; NA	S1 and S2: Desire to Use Drugs Inventory (DUDI)	S1: Oral fluoxetine 0, 20, or 40 mg daily or placebo; NA. S2: oral fluoxetine 20 mg or placebo, in addition to and 65 to 80 mg of methadone daily; individual therapy sessions	S1: DUDI: scores being highest at intake and diminishing during treatment (F = 10.9, df = 3147, *p* < 0.01). Fluoxetine produced no significant effect on “desire to use drug” over time by dose (f ratio < 1.0); S2: DUDI: Fluoxetine produced no significant decrease in craving in any medication group	S1: 55% (85); S2: 43% (10); NA
Harris et al., 2004, USA [28]	29; 40; 20:2; NA; M; 11 weeks	Cocaine dependence (DSM-IV); NA; NA	Within Session Rating Scale (VAS 100 mm: ‘‘desire to use’’ and ‘‘likely to use’’)	Oral fluoxetine 40 mg or placebo daily for 11 weeks; NA	VAS: Desire to use: no significant difference between groups; Likely to use: significant increase (*p* < 0.03) for de fluoxetine group	24% (7); N
Kolar et al., 1992, USA [29]	22; 35; 19:3; NA; O; 12 weeks	Cocaine dependence in methadone maintenance clients (DSM-III-R); S, IV; C	Self-report craving (0=“Not at all” to 20= “More than ever” over past 7 days)	Oral desipramine 200 mg or amantadine 200 mg or placebo for 12 weeks; weekly counseling sessions	Craving: significantly reduced over time in all three groups. However, there were no differences between groups on these measures (Time effect: F18.519 df 12/228 *p* <0.01; Group effect: F 0.9321 df 2/19 *p* N.S.)	32% (7); NA
Kosten et al., 1992, USA [30]	94; 32; 45:49; NA; O; 12 weeks	Opioid and cocaine dependence (DSM-III-R); NA; NA	VAS (analogue scale, with scores ranging from 0 (no craving) to 20 (most craving in life))	Oral amantadine hydrochloride300 mg or desipramine hydrochloride 150 mg or placebo daily for 12 weeks; weekly group relapse prevention therapy	VAS: Craving showed little change over the course of this trial and no difference across medication groups	21% (20); NA
McDowell et. al, 2005, USA [31]	111; 36; 83:28; Y; O; 1-week single blind lead-in placebo + 12 weeks of intervention	Cocaine dependence and major depression or dysthymia (DSM-III-R); S, I, IV; NA	Self-report cocaine craving (TLFB)	Oral desipramine up to 300 mg or placebo daily for 12 weeks; weekly individual manual-guided relapse prevention therapy	TLFB: no significant differences between the treatment groups in self-reported cocaine craving	58% (64); Y
Nanni- Alvarado et al., 2021, Mexico [32]	100; 28; 100:0; NA; M;12 week	DSM-IV criteria for cocaine use disorder; NA; A	Cocaine Craving Questionnaire (CCQ-G)	Oral mirtazapine 30 mg/kg or plcebo daily for 12 weeks; NA	CCQ: group treated with mirtazapine showed a significant decrease in mean craving compared to the placebo group (*p* < 0.01)	36% (36); N
Oliveto et al., 1995, USA [33]	21; 33; 11:10; NA; O; 12 weeks	Opioid-dependent and at least one positive cocaine urine and/or reported use of at least 14 g of cocaine over the 3-month period immediately preceding; NA; C	VAS: intensity of their desire for cocaine at the time they completed the form as well as in the past week on a scale from 0 (none at all) to 24 (more than ever)	Oral buprenorphine 8 mg plus either oral desipramine 150 mg, amantadine 300 mg, or fluoxetine 60 mg daily for 12 weeks; weekly group relapse prevention	VAS: craving reduced, but did not differ between groups	58% (10); NA
Passos et al., 2005, Brazil [34]	210; 31; 194:16; NA; O; 10 weeks	Cocaine dependence (DSM-IV); NA; NA	VAS: rated by the subject from 0 cm (no craving) to 10 cm (maximum craving)	Oral nefazodone 300 mg or placebo daily for 10 weeks; weekly individual psychotherapy	VAS: average craving did not differ between groups: 4.13 9 (nefazodone) and 4.17 (placebo)	79% (166); NA
Shoptaw et al., 2008, USA [35]	70; 37; 59:11; Y; O; 16 weeks intervention + 4 weeks observation	Cocaine abuse or dependence (DSM-IV); S, I, IV, O; C;	Cocaine Craving Scale (CCS)	Oral bupropion 300 mg or placebo daily for 16 weeks; cognitive behavioral therapy for 16 weeks	CCS: Cocaine craving decreased in both groups during the trial, although there were no statistically significant differences (*p* = 0.419)	82% (58); Y
Winstanley et al., 2011, USA [36]	145; 38; 79: 66; Y; O; 3 weeks of methadone+ 18 weeks of intervention + 12 weeks observation	Cocaine dependence and opioid dependence in treatment with methadone (DSM-IV); NA; NA	VAS	Oral fluoxetine 60 mg or placebo daily for 18 weeks; psychosocial counseling (individual and group)	VAS: There were no statistically significant between-group differences over time for craving scores	35% (51); Y

S1: study 1; S2: study 2.

**Table 2 brainsci-12-01546-t002:** Characteristics and main findings of the studies included in the systematic review–Psychostimulants.

Author, Year, Country	N (Sample); Age (Mean); Gender (M:F); Search for Treatment (Y: Yes; N: No); Setting (O: Outpatient; I: Inpatient; M: Mixed); Follow-up	Dependence; Route of Cocaine use (S: Smoked, I: Inhaled, IV: Intravenous, O: Oral); Cocaine Use Status (C: Current Use; A: Abstinent)	Craving Assessment Instrument	Pharmacotherapy and Duration; Non-Pharmacological Intervention	Outcomes (AG: Active Group; PG: Control/Placebo Group)	Dropout %(Number); Intention to Treat (ITT) Analysis (Y: Yes; N: No)
Alim et al., 1995, USA [37]	60; 33; NA; NA; I; 1-week baseline + 2 weeks of intervention	Cocaine dependence (DSM-III-R); S; NA	Questionnaire of Cocaine Urges (QCU); VAS	Oral diethylpropion 25, 50, 75 mg or 75 mg sustained release or placebo daily for 2 weeks; therapy	VAS: decrease in craving from baseline to end of intervention: F (1,45) = 27,39, *p* < 0.001; but no significant group differences; QCU: scores were higher in baseline than in the first 2 weeks of trial, but with no interaction with the groups	16,6% (10); NA
Anderson et al., 2009, USA [38]	210; 42; 148:59; Y; O; 12 weeks of intervention + 4 weeks of observation	Cocaine dependence (DSM-IV) NA; C	Brief Substance Craving Scale (BSCS); Cocaine Craving Questionnaire (CCQ)	Oral modafinil 200 mg or 400 mg or placebo daily for 12 weeks; one hour of individual weekly cognitive behavioral therapy	BSCS: craving declined on average within each active arm, and this difference from placebo was nominally significant for 200 mg (*p* = 0.04) but not for 400 mg (*p* = 0.90); CCQ: No significant effect on total score was noted, however, the sub-domains “Anticipation” (*p* = 0.04) and “Relief” (*p* = 0.03) were significant with treatment of 200 mg modafinil.	40% (75); NA
Buydens-Branchey et al., 1997, USA [39]	31; 37; 31:0; Y; I; 2 days of intervention + 2 days interval	cocaine dependence (DSM-III-R) S, I; A	VAS (chances they would use cocaine if they were not in the hospital and had unlimited financial resources. They were asked to draw a mark on a 100-mm line where 0 indicated that the chances were nil and 100 indicated that they were certain they would use cocaine)	Oral metachlorophenylpiperazine (m-CPP) 0.5 mg/kg of body weight or placebo for 2 days (separated by 48 h); NA	Craving: Patients’ craving for cocaine was found to decrease significantly (20%) after the administration of m-CPP (drug effect (F [1,1 = 41 2 5.4; *p* < 0.001))	NA; NA
Cantilena et al., 2012, USA [40]	20; 40; 18:2; N; I; 2 weeks intervention + observation after 2 weeks	Cocaine abuse or dependence (DSM-IV); NA; C	Brief Substance Craving Scale (BSCS)	Oral atomoxetine up to 100 mg or placebo daily for 2 weeks; NA	BSCS: craving reduced over time, but there were no significant differences between the atomoxetine and placebo groups on any days of assessment	20% (4); NA
Cornish et al., 2001, USA [41]	80; 38; 80:0; Y; O; 2 weeks placebo lead in + 4 weeks intervention + 4 weeks observation	Cocaine dependence (DSM-IIIR); NA; C	VAS (100-mm line their maximum level of craving over the previous 48-h period)	Oral ritanserin 10 mg or placebo daily for 4 weeks; counseling	VAS: neither the main effects nor the time by group interaction approached significance	24% (19); NA
Dackis et al., 2003, USA [42]	10; 44; 10:0; NA; O; 4 days	Cocaine dependence (DSM-IV); S; C	VAS (cocaine craving)	Oral modafinil 200 mg, or 400 mg or placebo daily for 4 days; NA	VAS: within each period, there was not a significant treatment effect	30% (3); NA
Dackis et al., 2005, USA [43]	62; 44; 44:16; Y; O; 8 weeks intervention + 1-week observation;	Cocaine dependence (DSM-IV); S; C	Brief Substance Craving Scale (BSCS); Cocaine Craving Questionnaire (CCQ)	Oral modafinil 400 mg or placebo daily for 8 weeks; twice-weekly cognitive behavioral therapy	BCSC, CCQ: no significant treatment group differences in the BCSC, or in the CCQ total score (Z-score¼0.76, *p*¼0.45)	8% (5); Y
Dackis et al., 2012, USA [44]	210; 45; 157:53; Y; O; 8 weeks	Cocaine dependence (DSM-IV); NA; C	Brief Substance Craving Scale (BSCS)	Oral modafinil 200 mg or 400 mg or placebo daily for 8 weeks; once-weekly cognitive behavioral therapy	BSCS: no treatment group differences	11,4% (24); Y
Ehrman et al., 1996, USA [45]	45; 38; 45:0; Y; O; 2 weeks screening + 4 weeks intervention	Cocaine dependence (DSM-III); S, I; C	VAS: scale of O-10	Oral ritanserin 10 mg or placebo daily for 4 weeks; NA	VAS: ritanserin had no effect on cue-induced craving	NA; NA
Goudriaan et. al, 2013, Netherlands [46]	29; 39; NA; Y; O; single-dose+ 1-week observation	Cocaine dependence (DSM-IV); NA; A	Cocaine Urge Questionnaire (CUQ)	Single dose of modafinil 200 mg or placebo; NA	CUQ: no significant changes in craving scores in the cocaine dependent were present in the modafinil versus the placebo condition (*p* = 0.39)	NA; Y
Grabowski et al., 1997, USA [47]	49; 34; 38:11; NA; O; 2-week intake period + 11-week medication trial+ 2-week observation	Cocaine dependence (DSM-IIIR); S, I; NA;	Self -report craving	Oral methylphenidate 45 mg or placebo daily for 11 weeks; therapy sessions 1 h/week	No significant differences between the two groups	51% (25); NA
Greenwald et al., 2010, USA [48]	13; NA; 6:2; N; I; 3 weeks	Opioid dependence and cocaine dependence (DSM-IV); S; C	Cocaine Craving Questionnaire (CCQ); VAS (0–100: Want Drug Again)	Oral d-amphetamine 30 or 60 mg sustained release (SR-AMP) or placebo daily for 3 days, followed by 4 conditions: cocaine (COC 100 mg+ saline), hydromorphone (COC 4 mg +HYD 24 mg), ‘speedball’ (COC 100 mg +HYD 24 mg) and placebo (COC 4 mg + saline); NA	CCQ: IG presented no significant reduction relative to PG (F 3.41 *p* (0.07)); VAS (Want drug again): IG presented a significant reduction relative to PG (F 8.48 *p* (0.01)	38% (5); N
Johns et al., 2021, USA [49]	25; 51; 11:2; N; M; 19 days	DSM-5 criteria for cocaine use disorder; S or IV; C	Visual Analogue Scale (VAS)	Oral lorcaserin 20 mg or placebo daily for 19 days; NA	VAS: there was a decrease in “I am craving” (*p*, 0.0001) for lorcaserin	48% (12); N
Johnson et al., 1997, USA [50]	65; 36; 55:10; Y; O; 2 weeks single-blind+ 4 weeks intervention+ 4 weeks observation	Cocaine dependence (DSM-IIIR); NA; NA	VAS	Oral ritanserin 10 mg or placebo daily for 4 weeks; psychoeducational program	VAS: reduction in craving in 66.4% and 32.5% for the placebo and ritanserin groups, respectively at end of intervention	27% (18); Y
Kampman et al., 2015, USA [51]	94; 47; 76:18; Y; O; 8 weeks	Cocaine dependence (DSM-IV); S; C	Brief Substance Craving Scale (BSCS)	Oral modafinil 300 mg or placebo daily for 8 weeks; weekly individual cognitive behavioral therapy	BSCS: placebo group was significantly more likely to report higher levels of craving than the modafinil group for intensity (OR = 2.04, 95% Ci = (1.06, 3.92), *p* = 0.03) and duration (OR = 1.89, 95% Ci = (1.06, 3.38), *p* = 0.03); there was a similar effect for frequency (OR = 1.51, 95% Ci = (0.84, 2.73)	24% (23); Y
Karila et. Al, 2016, France [52]	27; 38; 27:0; Y; I; 12 weeks	Cocaine dependence (DSM-IV); NA; C	Cocaine Craving Questionnaire (10-item CCQ brief)	Oral modafinil 400 mg/day for 26 days, then 300 mg/day for 30 days, and 200 mg/day for 31 days or placebo; NA	CCQ: No therapeutic advantage of modafinil 400 mg/day was detected during hospitalization	18% (5); NA
Levin et al., 2007, USA [53]	106; 37; 88:18; Y; O; 14 weeks	Cocaine dependence and attention deficit hyperactivity disorder (ADHD) (DSM-IV); S, I; C	VAS (craving during the last 24 h: “1” (very little) to “100” (very much))	Oral sustained-release methylphenidate 40 mg or placebo for 14 weeks; weekly individual cognitive behavioral therapy	VAS: Although both groups significantly reduced their craving severity over time (Z = −5.11, *p* < 0.001), there were no significant group or group by time differences	56% (59); Y
Mooney et al., 2009, USA [54]	82; 36; 54:28; Y; O; 8 weeks	Cocaine dependence (DSM-IV); NA; C	Cocaine craving scale (CCS)	Oral immediate release (IR) methamphetamine 30 mg or sustained release (SR)methamphetamine 30 mg or placebo daily for 8 weeks; 4 weeks counseling followed by 4 weeks of counseling plus a contingency management procedure	CCS: SR group reported less craving than the placebo group, t (36.3) = 2.49, *p* = 0.0451	68% (83); Y
Mooney et al., 2015, USA [55]	43; 45; 35:8; Y; O; 14 weeks	Cocaine dependence (DSM-IV); NA; C	VAS	Oral lisdexamphetamine (LDX) 70 mg or placebo daily for 14 weeks; weekly manual-based, cognitive-behavioral therapy (CBT)	VAS: those receiving LDX reported significantly less craving for cocaine than participants receiving placebo F(1, 62.6) = 5.94, *p* = 0.0176, (Placebo, M = 28.7, SE = 3.21; LDX, M = 17.5, SE = 3.30)	36% (16); Y
Nuijten et al., 2016, Netherlands [56]	73; 49; 66:7; NA; O; 12 weeks	Cocaine dependence (DSM-IV) and heroin assisted treatment; S; C	VAS (0–20)	Oral sustained-release dexamphetamine 60 mg or placebo daily, in addition to co-prescribed methadone and diacetylmorphine (max 150 mg) for 12 weeks; NA	VAS: significant changes in craving from baseline (Wald = 52·36; *p* < 0·001), but no significant group differences (Wald = 6·52; *p* = 0·011)	11% (8); Y
Perry et al., 2004, USA [57]	24; 38; 23:1; NA; O; 6 weeks	Schizophrenia or schizoaffective disorder and cocaine abuse or dependence (DSM-III); NA; NA	VAS	Oral mazindol (MZD) 6 mg or placebo (PBO) daily, in addition to their current antipsychotic medication for 6 weeks; cognitive-behavioral therapy	No significant effects were observed in craving: Baseline (Mean (SD)): PBO: 25.38 (29.12) MZD: 28.64 (29.98); Endpoint: PBO:32.14 (35.69) MZD: 13.17 (30.33)	16,6% (4); NA
Pirtle et al., 2019, USA [58]	11; 41; 11:0; N; M; 5 days of procedure, health status check 3 days after discharge	Cocaine Use Disorder (DSM-V); S, IV; C	VAS: 100 mm scroll bar marked from 0 (‘not at all’) to 100 (‘extremely) to indicate: ‘How much do you want to use cocaine?’	Two doses of oral lorcaserin 10 mg or placebo, separated by 24 h, followed by intravenous cocaine or placebo; NA	Lorcaserin treatment decreased craving following intravenous placebo [F(1, 214) = 15.8, *p* < 0.001] but not after cocaine	18% (2); N
Roache et al., 2000, USA [59]	S1: 57, 36, 45:12, NA, 11 weeks; S2:12, 37, 10:2, O, 1 week	Cocaine dependence (self-report); NA; C	VAS: 100-mm lines ranging from “not at all” to “extremely” for “am craving cocaine,” “desire to use cocaine,” “would use cocaine,” and “want to buy cocaine.”	S1: Oral methylphenidate 45 mg daily dose for 11 weeks; NAS2: Oral methylphenidate (15, 30, and 60 mg) followed by placebo for 1 week; NA	. S1: found neither positive nor negative influences of methylphenidate on treatment outcome; S2: found neither positive nor negative influences of methylphenidate on treatment outcome	.S1: 58% (33);.S2: NA; NA;
Santos et al., 2021, USA [60]	22; 39; 22:0; Y; O; 12 weeks	DSM-5 criteria for mild to severe cocaine use disorder; S and I; C	Visual Analogue Scale (VAS)	Oral extended release lorcaserin 20 mg or placebo daily for 12 week; NA	VAS: significant treatment effects in craving (*p*> 0.09)	14% (3); Y
Schubiner et al., 2002, USA [61]	48; 36; 43:5; Y; O; 12 weeks	Attention-deficit/hyperactivity disorder (ADHD) and cocaine dependence (DSM-IV); NA; NA	Cocaine Craving Questionnaire (CCQ)	Oral methylphenidate up to 90 mg or placebo daily for 12 weeks; twice-weekly cognitive-behavioral group therapy (CBT)	CCQ: there were no group differences in cocaine craving	47% (23); NA
Shearer et al., 2003, Australia [62]	30; 28; 16:14; NA; O; 14 weeks	cocaine dependence (DSM-IV); IV; C	VAS	Oral dexamphetamine 60 mg or placebo daily for 14 weeks; NA	VAS: no significant between-group differences; within-group changes were in favor of the treatment group and reached significance for cocaine craving (*p* < 0.01)	64% (19); Y
Stine et al., 1995, USA [63]	43; 35; 37:6; NA; O; 6 weeks	Cocaine dependence (DSM-IIIR); NA; C	VAS (5-point analog scale)	Oral mazindol 2 mg or placebo daily for 6 weeks; weekly group counseling	VAS: craving was not significantly affected by mazindol (drug: df = 1,41; F = 1.8; *p* = 0.2, time: df = 6,41; F = 10.4; *p* = 0.001, drug/time interaction: df = 6,41; F = 0.7; *p* = 0.6) although a decrease with time is observed	58% (25); NA
Walsh et al., 2013, USA [64]	50; 43; 36:14; Y; O; 12 weeks of treatment+ follow up after 12 weeks	Cocaine dependence (DSM-IV); S; C	Cocaine Craving Scale (CCS)	Oral atomoxetine 80 mg or placebo daily for 12 weeks; weekly counseling + cognitive behavioral therapy	CCS: During the trial, there was a significant group×time interaction (F(11,310) = 1.99; *p* = 0.03) for “All I want to use now is cocaine” and for “Nothing would be better than using coke right now” (F= 2.37; *p*= 0.008). Examination of the data suggest that these findings were not due to systematic changes over time in craving for either group; rather the placebo group reported slightly higher scores reliably throughout the trial and for one week (Week 11) the atomoxetine group reported higher scores (creating the interaction) which declined the following week. No other differences on craving items were observed.	44% (22); Y

S1: study 1; S2: study 2.

**Table 3 brainsci-12-01546-t003:** Characteristics and main findings of the studies included in the systematic review–Antipsychotics.

Author, Year, Country	N (Sample); Age (Mean); Gender (M:F); Search for Treatment (Y: Yes; N: No); Setting (O: Outpatient; I: Inpatient; M: Mixed); Follow-up	Dependence; Route of Cocaine Use (S: Smoked, I: Inhaled, IV: Intravenous, O: Oral); Cocaine Use Status (C: Current Use; A: Abstinent)	Craving Assessment Instrument	Pharmacotherapy and Duration; Non-Pharmacological Intervention	Outcomes (AG: Active Group; PG: Control/Placebo Group)	Dropout %(Number); Intention to Treat (ITT) Analysis (Y: Yes; N: No)
Akerele et al., 2007, USA [65]	28; 36; 25:3; NA; O; 14-weeks	Current cocaine and/or marijuana abuse or dependence and schizophrenia or schizoaffective disorder (DSM-IV); S, I; C	Cocaine Craving Report (CCR)	Oral olanzapine 5–20 mg or oral risperidone 3–9 mg daily for 10 weeks; weekly psychotherapy	CCR: there were no significant differences between the groups in terms of cocaine craving over time	43% (12); NA
Beresford et al., 2017, USA [66]	44; 48; 31:13; Y; O; 8 weeks of intervention + 2 weeks observation;	Cocaine dependence and schizophrenia (DSM-IV); NA; C	Brief Substance Craving Scale (BSCS)	Oral aripiprazole 10–30 mg or perphenazine 12–24 mg daily for 8 weeks; subjects were given the option to attend group behavioral therapy session weekly	BSCS: Contrasting weeks 3 to 5 vs 6 to 8 revealed significant late reductions in craving with aripiprazole. On the respective 5 points subscales, craving intensity decreased by 1.53 ± 0.43 (*p* < 0.0005) points, craving frequency by 1.4 ± 0.40 (*p* > 0.0004) points, and craving duration by 1.76 ± 0.44 (*p* > 0.0001) points	NA; NA
Hamilton et al., 2009, USA [67]	48; 46; 48:0; Y; O; 16 weeks intervention + 4 weeks observation	Cocaine dependence (DSM-IV); S, I, IV, O; C	Craving Questionnaire	Oral olanzapine 2,5–20 mg or placebo daily for 16 weeks; psychotherapeutic and educational groups	Craving Questionnaire: significant within-subjects (time) effects, but there were no significant differences between the olanzapine and placebo groups	NA; NA
Kampman et al., 2003, USA [68]	30; 40; 22:8; Y; O; 12 weeks	Cocaine dependence (DSM-IV); S, I, IV; C	Brief Substance Craving Scale (BSCS)	Oral olanzapine 10 mg or placebo daily for 12 weeks; twice-weekly individual cognitive-behavioral	BSCS: significant decline over the 12 weeks: duration (t//3.16, *p*/0.002); frequency (t//2.94, *p*/0.004), and intensity (t//2.93, *p*/0.004). There was no medication effect on either duration (t//0.04, *p*/0.97), frequency (t/0.50, *p*/0.62), or intensity (t//0.39, *p*/0.7)	10% (3); NA
Loebl et al., 2008, USA [69]	31; 43; 31:0; NA; O; 12 weeks	Cocaine dependence (DSM-IV); NA; C	Minnesota Cocaine Craving Scale (MCCS)	Intramuscular risperidone 25 mg or placebo every other week for 12 weeks; NA	MCCS: craving reduced in both groups, but with no difference between them	NA; NA
Lofwall et al., 2014, USA [70]	34; 35; 19:2; N; I; 32 days	Self-report use; S; C	VAS: “Desire for Cocaine”	Oral aripiprazole 2 or 10 mg or placebo daily for 32 days; NA	No significant group effects were observed	40% (13); NA
Moran et al., 2017, USA [71]	18; 45; 17:1; Y; O; 12 weeks intervention + 29 weeks of buprenorphine treatment	Dependence on opioids, current cocaine use on at least 3 of the last 30 days, lifetime cocaine-use duration of at least one year; S, I, IV; C;	Ecological momentary assessment (EMA): participants answered craving questions with the response options “NO!!”, “no??”, “yes??”, and “YES!!”	Oral aripiprazole 15 mg or placebo daily for 12 weeks; weekly session of individual counseling	EMA: in randomly prompted EMA entries craving was not reported frequently in either group but tended to be reported more often in the aripiprazole group, F (1,13) = 2.91, *p* =0.11, effect = 0.43, 95% CI = −.08–0.76. Almost all participants in the aripiprazole group reported craving at least once	22% (4); Y
Reid, Casadonte et al., 2005, USA [72]	63; 39; 50:13; Y; O; 8 weeks	Cocaine dependence (DSM-IV); S, I, IV; C	Cocaine Craving Questionnaire general (CCQ-general); Brief Substance Craving Scale (BSCS)	Oral olanzapine 10 mg, valproate 1500 mg, coenzyme Q10 200 mg, and L-carnitine 500 mg combination or placebo daily for 8 weeks; cognitive behavioral counseling	CCQ: significant treatment effect (*p* < 0.05), in which the olanzapine group showed significantly higher craving than placebo at week 8 (*p* < 0.05); BSCS: scores for cocaine craving decreased in all groups by the end of treatment; however, olanzapine group experienced a weaker reduction in cocaine craving at the last clinic visit (*p* < 0.05);	42% (29); NA;
Sayers et al., 2005, USA [73]	24; 46; 23:1; NA; O; 26 weeks	Schizophrenia and cocaine addiction (DSM-IV); NA; C	VAS: mark on a 100-mm line representing the “greatest degree of craving for cocaine since your last visit,” with endpoints marked as “NONE” and “MORE THAN EVER.”	Oral olanzapine 10 mg or haloperidol 10 mg daily for 26 weeks; NA	VAS: significant difference in craving over time favoring the patients in the haloperidol group	42% (10); Y
Smelson et al., 2004, USA [74]	35; 41; NA; Y; O; 2 weeks	Cocaine dependence (DSM-IV); NA; C	Voris Cocaine Craving Questionnaire (VCCQ)	Oral risperidone 1 mg or placebo daily for 2 weeks; NA	VCCQ: significant main effect of time for the craving (F = 33.62, *p* = 0.01). The interaction of treatment group by time, however, was not significant for the craving	8% (3); NA
Smelson et al., 2006, USA [75]	31; 43; NA; NA; O; 6 weeks	Cocaine-dependent patients with schizophrenia (DSM-IV); NA; C	Voris Cocaine Craving Questionnaire (VCCQ)	Oral olanzapine 10 mg or haloperidol 10 mg daily for 6 weeks; enhancement therapy, relapse prevention, psychoeducational skills training, and a 12-step therapy	VCCQ: olanzapine-treated subjects compared with those in the haloperidol group showed significantly less cue-elicited craving: Energy score (M = 39.1, SD = 9.2 vs. M = 27.6, SD = 12.8), t(16) = 2.20, *p* = 0.04 (2-tailed), d = 0.99, but not on the Intensity (M = 8.5, SD = 5.7 vs. M = 14.4, SD = 11.8), t(16) = 1.39, *p* = 0.18 (2-tailed), d = 0.64	42% (13); NA
Tapp et al., 2015, USA [76]	60; 48; 52:8; NA; O; 12 weeks	Cocaine dependence (DSM-IV); NA; C	Brief Substance Craving Scale (BSCS)	Oral quetiapine 400 mg or placebo daily for 12 weeks; weekly cognitive-behavioral therapy group session	BSCS: did not differ in terms of absence of cravings (34.5% in quetiapine group versus 29.0% in placebo group; Wald statistic = 0.21, df = 1, *p* = 0.65)	68% (41); Y

**Table 4 brainsci-12-01546-t004:** Characteristics and main findings of the studies included in the systematic review–Anticonvulsants.

Author, Year, Country	N (Sample); Age (MEAN); Gender (M:F); Search for Treatment (Y: Yes; N: No); Setting (O: Outpatient; I: Inpatient; M: Mixed); Follow-up	Dependence; Route of Cocaine Use (S: Smoked, I: Inhaled, IV: Intravenous, O: Oral); Cocaine Use Status (C: Current Use; A: Abstinent)	Craving Assessment Instrument	Pharmacotherapy and Duration; Non-Pharmacological Intervention	Outcomes (AG: Active Group; PG: Control/Placebo group)	Dropout %(Number); Intention to Treat (ITT) Analysis (Y: Yes; N: No)
Bisaga et al., 2006, USA [77]	99; 39; 87:12; NA; O; 2 weeks single-blind placebo lead-in period + 12 weeks intervention +2 weeks lead-out	Cocaine dependence (DSM-IV); S, I; C	Self-reported number of days in which participants experienced cocaine craving	Oral gabapentin 3200 mg or placebo daily for 12 weeks; weekly individual relapse-prevention therapy	Craving: participants observed a decrease in craving frequency (z = −2.09, 0.04), but there were no significant effects of treatment or level of cocaine use	51% (50); Y
Brown et al., 2012, USA [78]	112; 44; 67:45; NA; O; 10 weeks	Cocaine dependence (DSM-IV); S, I, IV; C	Cocaine-Craving Questionnaire (CCQ)	Oral lamotrigine 200–400 mg or placebo daily for 10 weeks; NA	CCQ: no differences between groups (*p* = 0.53)	35% (42); Y
Crosby et al., 1996, USA [79]	44; 34; 35:9; Y; O; 12 weeks	Cocaine abuse or dependence (DSM-III-R); S, I, IV; C	Minnesota Cocaine Craving Scale (MCCS)	Oral phenytoin 300 mg or placebo daily for 12 weeks; weekly counseling	MCCS: difference did not approach significance (est = −4.96; SE = 5.69; *p* = 0.383)	73% (32); NA
Halikas et al., 1997, USA [80]	150; 33; 106:44; Y; O; 12 weeks intervention	Cocaine dependence (DSM-III-R); NA; C	Minnesota Cocaine Craving Scale (MCCS)	Oral carbamazepine 400 mg or 800 mg or placebo daily for 12 weeks; psychosocial treatment programming	MCCS: compared with placebo, the 400 mg treatment condition exhibited a reduction in intensity and duration of craving over the course of the study (est= −0.113; p< 0.001)	62% (93); Y
Johnson et al., 2013, USA [81]	24; 34; 19:5; N; I; 10 days intervention + 6 days experiments + 7 days interval	Cocaine dependence (DSM-IV); NA; C	VAS (100 mm left to right “not at all” to “extremely”)- (“crave”, “desire”, “want cocaine”, and “could refuse cocaine”)	Oral topiramate 200 mg or placebo daily for 5 days followed by 3 days of cocaine IV experiments, repeated after 7 days; NA	Bidirectional effect: wherein the effects of the highest cocaine dose (0.65 mg/kg) on VAS-Craving was decreased by topiramate, but not with low cocaine dose (0.325 mg/kg)	NA; NA
Johnson et al., 2013 USA, [82]	142; 44; 103:39; NA; O; 12 weeks;	Cocaine dependence (DSM-IV); S, I; C	Brief Substance Craving Scale (BSCS)	Oral topiramate up to 300 mg or placebo daily for 12 weeks; weekly cognitive-behavioral treatment	BSCS: topiramate vs placebo were 0.499 vs 0.300 (OR, 2.33; 95% CI, 1.15–4.71; *p* = 0.02) for having “reportedly no craving at all” in terms of the intensity, frequency, and duration of craving in the past 24 h and 0.501 vs 0.271 (OR, 2.70; 95% CI, 1.38–5.29; *p* = 0.004) for having “reportedly no craving at all” in the intensity of craving on the worst day	49% (70); Y
Kampman et al., 2013, USA [83]	170; 44; 135:35; Y; O; 1-week baseline + 12 weeks intervention	Cocaine and alcohol dependence (DSM-IV); S; C	Minnesota Cocaine Craving Scale (MCCS)	Oral topiramate 300 mg or placebo daily for 12 weeks; weekly individual psychotherapy (CBT)	MCCS: declined significantly during the trial in both groups and there were no between-group differences	42% (70); Y
Reid et al., 2009, USA [84]	20; 44; 16:4; NA; O; 8 days intervention + 11 days observation	Cocaine dependence (DSM-IV); NA; C	Brief Substance Craving Scale (BSCS); VAS: with descriptors “not at all”, “mildly”, “moderately” and “extremely” equally spaced above line from 0 to 100: “how much do you desire to use cocaine right now?”	Oral valproate up to 1500 mg or placebo daily for 8 days; NA	VAS: “desire to use cocaine now” (F(1,38) = 3.916, *p* < 0.05), in which cocaine cue-induced craving levels were higher in the valproate condition. BSCS: slightly more cocaine craving (BSCS total score) in the valproate condition (F(1,38) = 2.326, *p* = 0.103)	15% (3); NA
Somoza et al., 2013, USA [85]	186; 45; 125:61; Y; O; 12 weeks	Cocaine dependence (DSM-IV); S, I, IV; C	Brief Substance Craving Scale (BSCS)	Oral vigabatrin 3 g or placebo daily for 12 weeks; weekly computerized cognitive behavioral therapy and biweekly individual counseling for 13 weeks	BSCS: both groups reported less craving, but there was no significant medication effect	27% (51); Y
Umbricht et al., 2014, USA [86]	171; 42; 89:82; Y; O; 12 weeks	Cocaine dependent methadone maintenance patients (DSM-IV); NA; NA	Cocaine Selective Severity Assessment (CSSA)	Oral topiramate up to 300 mg or placebo daily + methadone treatment (median 100 mg/day); NA	CSSA: there was no effect of topiramate on craving scores over time	34% (58); Y
Winhusen, Somoza, Ciraulo et al., 2007, USA [87]	141; 42; 134: 7; Y; O; 12 weeks	Cocaine dependence (DSM-IV); NA; C	Brief Substance Craving Scale (BSCS)	Oral tiagabine 20 mg or placebo daily for 12 weeks; individual cognitive behavioral therapy on a weekly	BSCS: both groups reported significantly less craving over the course of the study, but with no significant difference between the groups as indicated by the non-significant medication by week (Z = 0.09, *p* > 0.05) and medication (Z = 1.68, *p* > 0.05) effects	44% (62); NA

**Table 5 brainsci-12-01546-t005:** Characteristics and main findings of the studies included in the systematic review–Other Drugs.

Author, Year, Country	N (sample); Age (Mean); Gender (M:F); Search for Treatment (Y: Yes; N: No); Setting (O: Outpatient; I: Inpatient; M: Mixed); Follow-up	Dependence; Route of Cocaine Use (S: Smoked, I: Inhaled, IV: Intravenous, O: Oral); Cocaine Use Status (C: Current Use; A: Abstinent)	Craving Assessment Instrument	Pharmacotherapy and Duration; Non-Pharmacological Intervention	Outcomes (AG: Active Group; PG: Control/Placebo Group)	Dropout % (Number); Intention to Treat (ITT) Analysis (Y: Yes; N: No)
Becker et al., 2020, USA [88]	36; 42; 28:8; Y; O; single dose + 4 weeks of observation	Cocaine dependence (DSM-IV); NA; C	Craving Questionnaire (CCQ-10)	Single dose of intravenous lidocaine 2 mg/kg immediately following a cocaine craving script (lidocaine/ craving), saline following a craving script (saline/craving), or lidocaine following a relaxation script (lidocaine/relax); weekly cognitive behavioral therapy	CCQ: craving appeared higher in the lidocaine/craving and lidocaine/relax groups relative to saline/craving, but there were no significant differences between treatment groups in craving (Mean scores: Baseline: lidocaine/ craving: 31.7 (±12.6) saline/craving: 17.3 (±9.5 lidocaine/relax: 22.7 (±11.9);Endpoint: lidocaine/ craving: 32.3 (±18.5)saline/craving: 15.8 (±4.8)lidocaine/relax: 20.9 (±15.6))	14% (5); NA
Bisaga et al., 2010, USA [89]	81; 40; 63: 17; NA; O; 2 weeks of placebo lead-in + 12 weeks intervention + 2 weeks placebo	Cocaine dependence (DSM-IV); NA; C	Self-report of weekly proportion of days with craving	Oral memantine 40 mg or placebo daily for 12 weeks; individual relapse-prevention therapy	Craving: no changes between the two treatment groups (X2(1) = 0.41, *p* = 0.52) or over time (X2(1) = 0.01, *p* = 0.91)	40% (32); Y
Brown et al., 2015, USA [90]	122; 42; 82:40; NA; O; 12 weeks	Cocaine dependence and co-occurring bipolar I disorder (depressed or mixed mood state) (DSM-IV) S, I, IV; C	Cocaine Craving Questionnaire (CCQ)	Oral citicoline up to 2000 mg or placebo daily for 12 weeks; therapy	CCQ: no significant group or group-by-time effects were observed (Treatment group F 2.4 df 1, 108 *p* 0.127; treatment group by time F 1.3 df 1, 101 *p* 0.249)	34% (44); Y
Chadwick et al., 1990, USA [91]	50; 30; 25:4; NA; I; 4 weeks intervention;	Cocaine dependence (DSM-III) S, I, IV; C	VAS (craving scored on a 0 to 10 scale, with 0 indicating no craving and 10 indicating the worst craving ever felt)	Oral L-tryptophan 1 g and L-tyrosine 1 g or placebo for 2 weeks, then 2 weeks crossed over; NA	VAS: amino acids did not significantly reduce drug craving	42% (21); NA
Dackis et al., 1987, USA [92]	13; 28; 9:4; NA; I; 2 days	Cocaine abuse (DSM-III); S, I, IV; C	VAS: 100 mm scale (the right pole indicating maximal craving, and the left pole indicating no craving)	Single dose of bromocriptine 1,25 mg or placebo; NA	VAS: the mean (± SD) R value after bromocriptine (16.2 ± 14.7) was significantly greater (t = 1.84, df= 12, *p* < 0.05) than after placebo (10.2 ± 13.0)	NA; NA
Dakwar, Levin et al., 2014, USA [93]	11; 47; 7:1; N; I; 9 days intervention+ 4 weeks observation	Cocaine dependence (DSM-IV); S; C	VAS: 100 mm scale corresponding to the intensity of their wanting cocaine, from “none at all” on the left to “extremely” on the right	Intravenous ketamine (0.41 mg/kg or 0.71 mg/kg), administered at three doses, during 52-min or lorazepam 2 mg, separated by 48 h; 10-min mindfulness-based exercise	VAS: reduction in cue-induced craving (*p* = 0.012) in ketamine group relative to lorazepam group	27% (3); N
Dakwar, Anerella et al., 2014, USA [94]	8; 48; NA; N; I; 9 days intervention + 4 weeks observation	Cocaine dependence (DSM-IV); S; C	VAS: 100 mm scale corresponding to the intensity of their wanting cocaine, from “none at all” on the left to “extremely” on the right	Intravenous ketamine (0.41 mg/kg or 0.71 mg/kg), administered at three doses, during 52-min or intravenous lorazepam 2 mg, separated by 48 h; NA	VAS: mystical-type effects did not mediate cue-induced craving	0; NA
Dakwar et al., 2017, USA [95]	20; 49; 11:9; N; I; 3 days intervention + 7 weeks observation	Cocaine dependence (DSM-IV); NA; NA	VAS	Intravenous ketamine (0.71 mg kg^−1^), administered at three doses over 6 days or of the active control midazolam (0.025 mg kg); NA	VAS: craving significantly reduced initially but not throughout the monitoring period; ketamine led to significant craving reduction (%) prior to discharge (59.6 vs 15.3%, t17 df = 3.44, *p* < 0.01) but not at subsequent time-points	NA; NA
Dieckmann et al., 2014, Brazil [96]	111; 32; 111:0; NA; O; 8 weeks	Cocaine dependence (DSM-IV); S, I; C	Minnesota Cocaine Craving Scale (MCCS)	Oral biperiden 6 mg or placebo daily for 8 weeks; brief cognitive-behavioral psychotherapy	MCCS: reduction of 19.1% of craving in the placebo group (*p* = 0.017) and 37.6% in the biperiden group (p<0.001)	69% (77); Y
Eiler et al., 1995, USA [97]	63; 36; 63:0; Y; I; 18 days intervention	Cocaine dependence (DSM-III); S, I, IV; C	VAS: a single horizontal line which represents craving for cocaine on a continuum, starting with 0, representing no cocaine craving, up to 10, which represents the highest degree of cocaine craving	Oral bromocriptine up to 10 mg or placebo daily for 18 days; NA	VAS: time effect F = 3.46, *p* < 0.001; first week: time effect F = 3.84, *p* < 0.002). Bromocriptine did not seem to reduce cocaine craving more expeditiously or quantitatively than placebo.	53% (34); NA
Elkashef et al., 2006, USA [98]	300; 41; 234:66; Y; O; 2 weeks baseline+ 8 weeks intervention	Cocaine dependence (DSM-IV); S, I; C	Brief Substance Craving Scale (BSCS)	Selegiline Transdermal System (STS) 6 mg or placebo (PBO) patches daily for 8 weeks; individual psychotherapy weekly	BSCS: difference between the selegiline and placebo groups was not significant (*p* = 0.96): Mean weekly scores: baseline: PBO: 6.7 (2.7), STS: 6.5 (2.5); endpoint: PBO: 4.1 (3.0), STS: 4.5 (3.0)	31% (93); NA
Fox et al., 2012, USA [99]	29; 39; 19:10; Y; I; 4 weeks	Cocaine dependence (DSM-IV); NA; C	VAS: intensity of your desire to use cocaine/nicotine at the moment, in which 1 = ‘not at all’ and 10 = ‘extremely high’	Oral guanfacine up to 3 mg or placebo daily for 4 weeks; group drug counseling	VAS: guanfacine significantly decreased cue-related cocaine craving	NA; NA
Fox et al., 2013, USA [100]	42; 42; 24:18; Y; I; 1-week baseline + 1-week intervention + 2 weeks observation	Cocaine dependence (DSM-IV); NA; C	Cocaine Craving Questionnaire-Brief (CCQ)	Oral progesterone 400 mg or placebo daily for 7 days; 12-Step based Group Drug Counseling Manual	CCQ: progesterone group significantly presented lower levels of cocaine craving compared with placebo (*p* < 0.05) in both males and females	NA; NA
Handelsman et al., 1995, USA [101]	59; 36; 59:0; NA; O; 1-week single blind placebo+ 8 weeks intervention	Cocaine dependence in methadone-maintained treatment (DSM-IIIR); NA; C	VAS: average craving and peak craving for each day scored from zero (not at all) to ten (most ever)	Oral amantadine 200 mg or and 400 mg or placebo daily for 8 weeks; NA	VAS: no difference across medication groups for craving, but there was a reduction overall over time	13% (8); NA
Handelsman et al., 1997, USA [102]	50; 38; 50:0; Y; O; 5 weeks intervention + 3-month psychotherapy;	Cocaine dependence in methadone-maintained treatment (DSM-III-R-); NA; C	Cocaine Craving Questionnaires (CCQ); VAS (the average daily craving and perceived resistance to using cocaine, assessed by the use of aloo-mm, non-numerated visual analog scale anchored from 0 (not at all) to 100 (most ever))	Oral bromocriptine 2,5 mg or placebo daily for 5 weeks; cognitive behavioral therapy	CCQ and VAS: bromocriptine group did not differ from the placebo group in craving for cocaine	20% (10); NA
Jobes et al., 2011, USA [103]	59; 41; 50:9; N; O; 1 day	Self-report of using cocaine for at least 1 year and at least once in the previous 30 days; S, I; C	Cocaine Craving Questionnaire (CCQ); VAS (questions worded, respectively, “Right now, how much do you [crave/want/need] cocaine?”)	Single dose of oral clonidine 0.1 or 0.2 mg or placebo; NA	VAS: in the placebo group, Crave Cocaine was significantly increased from baseline after both the stress script and the drug-cue script, but not after the neutral script; in the 0.1-mg group, Crave Cocaine increased only after the drug-cue script, not the stress script; in the 0.2-mg group, Crave Cocaine did not increase after either active script. No significant effects on the VAS measure Need cocaine; CCQ: there was also a significant effect of clonidine dose, with a dose-related decrease in craving scores [F(2,56) = 5.49, *p* = 0.007]	0; NA
Jobes et al.; 2015; USA [104]	35; 42; 16:17; NA; M; 6 weeks	Cocaine-abusing outpatients who were also being maintained on methadone for heroin dependence; NA; C	Cocaine Craving Questionnaire (CCQ), VAS (was worded “Please rate the intensity of your desire to use cocaine AT THIS MOMENT”);	Intravenous propranolol 40 mg or dextrose placebo intravenously (3 interventions, 1 dose in each); NA	VAS: propranolol acutely increased reactivity to the cocaine; CCQ: increased scores in the propranolol group only	7% (2); N
Johnson et al., 2004, USA [105]	18; 33; 12:6; N; O; 4 weeks	Cocaine dependence (DSM-IV); NA; NA	VAS	Oral isradipine 15 or 30 mg or placebo; plus cocaine HCl 0.325 or 0.650 mg/kg or placebo, for 9 sessions separated by a 2-day interval; NA	VAS: no main isradipine effect in craving	NA; NA
Johnson et al., 2019, USA [106]	39; 51; 27:12; Y; O; 2 weeks of induction phase+ 3 weeks of intervention+ 2 weeks observation	Cocaine dependence (DSM-IV); I; NA	Cocaine Craving Questionnaire-Now (CCQ)	Oral D-cycloserine (DCS) 50 mg or placebo daily for 3 weeks; Contingency management (CM) intervention	CCQ: craving decreased for both groups following the introduction of CM and then for the DCS group, increased significantly during the posttreatment phase (post hoc pairwise comparison, *p* 0.01)	21% (11); N
Kablinger et al., 2012, USA [107]	26; 42; 17:9; Y; O; 6 weeks of interventions + 2 weeks of observations	Cocaine dependence (DSM-IV); NA; C	Cocaine Craving Questionnaire (CCQ)	Oral metyrapone/ oxazepam 500/20 mg or 1500/20 mg or placebo daily for 6 weeks; NA	CCQ: a change from baseline to endpoint did not reach statistical significance for the pooled group compared with placebo, but reached statistically significance in some visits	51% (23); NA
Kampman et al., 2001, USA [108]	108; 36; 88:20; Y; O; 1-week placebo lead in + 8-week intervention	Cocaine dependence (DSM-IV); S, I, IV; C	VAS	Oral propranolol up to 100 mg or placebo daily for 8 weeks; twice-weekly individual cognitive-behavioral	VAS: decline in cocaine craving during the trial (Wald x2= 24.0, *p* = 0.001) but there was no significant medication group by time interaction (Wald x2 = 7.4, *p* = 0.387)	47% (51); NA
Kampman et al., 2006, USA [109]	199; 41; 130:69; Y; O; 2-week baseline phase + 8-weeks of intervention	Cocaine dependence (DSM-IV); S, I, IV; C	Brief substance craving scale (BSCS)	Oral amantadine 300 mg or propranolol 100 mg or combination of amantadine 300 mg + propranolol 100 daily or placebo for 8 weeks; twice-weekly individual cognitive-behavioral therapy	BSCS: significant decline over the trial in all groups but there were no significant group effects or group by time interactions	41% (82); Y
Kampman et al., 2010, USA [108]	60; 45; 45:15; Y; O; 1-week baseline + 8 weeks intervention	Cocaine dependence (DSM-IV); S, I; C;	Brief Substance Craving Scale (BSCS)	Oral acamprosate 666 mg 3 times daily or placebo for 8 weeks; weekly individual cognitive behavioral relapse prevention therapy	BSCS: cocaine craving showed a significant decline over the trial in both groups (F = 19.89, *p* < 0.001), with no difference between groups	40% (24); NA
Larowe et al., 2013, USA [110]	126; 43; 83:28; Y; O; 8 weeks;	Cocaine dependence (DSM-IV); S; C;	Brief Substance Craving Scale (BSCS);	Oral N-acetylcysteine 1200 mg, 2400 mg or placebo daily for 8 weeks; weekly session of manual-based cognitive therapy	. BSCS: time-related reduction, Wald χ2= 104.1, df = 8, *p*< 0.001; no difference between groups (among abstinent participants, NAC presented significant less craving rates)	44% (55); N
Lee at. Al., 2015, USA [111]	23; 38; 22:1; NA; I; 1 week	Cocaine dependence (DSM-IV); S; A	(VAS): “If your drug of choice was in front of you right now, what’s the likelihood that you would use?”	Intranasal oxytocin (OT) 24 IU or placebo, for 4 sessions; NA	VAS: the effect of OT on desire to use before exposure to cues was small (mean ± SE: OT = 2.57 ± 0.39; PL = 1.91 ± 0.39) but significant (df = 1,22; F = 5.22, *p* = 0.032), where desire to use was augmented under OT	NA; NA
Leyton et al., 2005, USA [112]	8; NA; 8:0; N; M; 3 days	Nondependent, regular cocaine users (DSM-IV); NA; C	VAS items: want cocaine, crave cocaine, urge for cocaine, and desire cocaine −1 (least) and 10 (most)	Balanced amino acid mixture or acute phenylalanine–tyrosine depletion (APTD), followed by L-dopa/carbidopa (2 × 100 mg/25 mg) or placebo; NA	VAS: cue- and cocaine-induced craving was significantly reduced by APTD and APTD + L-dopa, for the following: want cocaine, F(10, 70) = 3.27, *p* < 0.002; urge for cocaine, F(10, 70) = 2.10, *p* < 0.04; and the total craving score, F(10, 70) = 2.25, *p* < 0.025	0; NA
Licata et al., 2011, USA [113]	29; 38; 18:11; N; O; 8 weeks intervention + 4 weeks observation	Cocaine dependence (DSM-IV); S, I; C	VAS: ranged from 0 ‘none at all’ to 10 ‘extremely high’ of their desire to use cocaine	Oral citicoline 1000 mg or placebo daily for 8 weeks; weekly group therapy	Cocaine craving decreased during the treatment period for both treatment groups, but not significantly	38% (11); NA
Malcolm et al., 2000, USA [114]	358; NA; 282:76; Y; O; 12 weeks intervention + 4 months follow up visits	Cocaine dependence and combined cocaine/alcohol dependence (DSM-III-R); NA; C	VAS	Oral pergolide 0.10 or 0.50 mg or placebo for 12 weeks; 12-step based group therapies	VAS: Although craving scores rose considerably over the first week of the study (range 40–50 mm), scores did not differ for any of the 12-week visits for any of the treatment groups	66% (202); Y
Malcolm et al., 2005, USA [115]	116; 36; 89:27; NA; O; 12 weeks	Cocaine dependence (DSM-IV); S, IV; C	Unclear	Oral amlodipine 5–10 mg or placebo daily for 12 weeks; 12 standard manual-driven cognitive behavioral therapy sessions	Craving score did not differ significantly between treatment arms	81% (90); NA
Meneses-Gaya et al., 2020, Brazil [116]	31; 32,9; 31:0; NA; I; 10 days	DSM-IV diagnosis of crack-cocaine dependence;	Cocaine Craving Questionnaire–Brief (CCQ-Brief); Minnesota Cocaine Craving Scale (MCCS);	Oral canabidiol (CBD) 300 mg or placebo daily for 10 days; group psychotherapy once a week	CCQ-Brief: significant effect of time (F10,230 = 16.174; *p* < 0.001), but not of treatment (F10,230 = 2.663; *p* = 0.116) or time/treatment interaction (F10,230 = 0.489; *p* = 0.897).MCCS: significant effect of time (F10,230 = 16.450; *p* < 0.001), but not of treatment (F10,230 = 2460	19% (6); Y
Milivojevic et al., 2016, USA [117]	46; 41; 29:17; Y; I; 1 week	Cocaine dependence (DSM-IV); NA; C	Cocaine Craving Questionnaire-Brief (CCQ)	Oral micronized progesterone 400 mg or placebo daily for 7 days; NA	CCQ: active group had significantly lower post-imagery cocaine craving	NA; NA
Modesto-Lowe, 1997, USA [118]	26; 36; 23:3; Y; O; 8 weeks	Comorbid alcohol and cocaine abuse/ dependence (DSM-IV); NA; NA	VAS: 5- point rating scale that range from ‘none at all’ to ‘extremely’-(‘How strong is your desire for cocaine right now?’); Cocaine Craving Questionnaire (CCQ)	Oral naltrexone 50 mg or placebo daily for 8 weeks; NA	VAS/CCQ: there was no effect of medication on the desire for cocaine	NA; NA
Mongeau-Pérusse et al., 2021, Canada [119]	78; 45,9; 64:14; NA; M; 12 weeks	DSM 5 criteria for cocaine use disorder; S, I and IV; A	VAS:ranging from 0 to 10; Cocaine Craving Questionnaire-Brief (CCQ-Brief); Cocaine Selective Severity Assessment (CSSA)	Oral cannabidiol (CBD) 800 mg or placebo daily for 12 weeks after 10 days of inpatient detoxification; relapse prevention group session	VAS, CCQ, CSSA: CBD did not reduce cocaine craving	37% (28); NA
Mooney et al., 2006, USA [120]	S1: 67, S2:122; S1:35, S2:39; S1: 45:22, S2: 103:19; NA; O; S1: 5 weeks, S2: 9 weeks	Cocaine users (DSM-IV); S; C	S1: VAS S2: Cocaine subscale of the Brief Substance Craving Scale (BSCS)	S1: L-dopa/carbidopa 800/200 mg or placebo daily for 5 weeks; supportive behavioral counseling; S2: L-dopa/carbidopa 800/200 mg or 1600/400 mg or placebo daily for 9 weeks; supportive behavioral counseling	S1: Craving: no differences between groups; S2: Craving: no differences between groups	S1: 62% (45), YS2: 57% (70); Y
Moran-Santa Maria et al., 2015, USA [121]	84; 41; 71:13; N; M; 3 days	Cocaine dependence (DSM-IV); NA; C	Cocaine Craving Questionnaire (CCQ)	Single dose of oral guanfacine 2 mg or placebo; NA	CCQ: participants in the no-stress group reported significantly greater craving in response to the cocaine-cue as compared to the neutral cue (*p* < 0.001), in both the no-stress guanfacine (*p* < 0.001) and no-stress placebo (*p* = 0.023) groups. In the stress group, participants in both the guanfacine and placebo groups reported similar craving ratings in response to the cocaine and the neutral cue (*p* = 0.480)	NA; Y
Nann-Vernotica et al., 2001, USA [122]	10; 38; 9:1; N; I; 1 week	Cocaine dependence (DSM-IV); S, IV; C	VAS: “how much do you desire cocaine?”	Four doses of oral ecopipam or placebo (0, 10, 25, and 100 mg) daily for 1 week; NA	VAS: Ecopipam largely failed to alter the desire for cocaine	NA; NA
Nasser et al., 2014, USA [123]	29; 40; 23:6; N; NA; 1-day intervention + 6 days observation	Cocaine abuse (DSM-IV); NA; C	Brief substance craving scale (BSCS); VAS	Single dose RBP-8000 (100 or 200 mg) or placebo; NA	BSCS: results were low and fairly steady over the entire assessment period; VAS: results were similar over the entire assessment period and across treatment groups	10% (3); NA
Newton at. Al., 2015, USA [124]	18; 40; 11:7; N; I; 15 days	Cocaine dependence (DSM-IV-TR); S, IV; NA	VAS: participants rated “DESIRE” cocaine, anchored at 0 (not at all) and 100 (most ever)	Oral pramipexole up to 3.0 mg or placebo daily for 15 days; NA	VAS: No significant main effects was found for “DESIRE”	0; Y
Petrakis et al., 2000, USA [125]	67; NA; 32:35; NA; O; 12 weeks	Cocaine dependence in methadone-maintained opioid-dependent (DSM-IIIR); S, I, IV; C	Self-report cocaine craving	Oral disulfiram 250 mg or placebo daily, was placed directly in the methadone for 12 weeks; weekly individual and group counseling	There was a significant decrease in craving over time for the entire sample (Z 5 9.05; *p*, 0.01), but no significant decrease in craving by treatment group	23% (13)
Plebani et al., 2012, USA [126]	37; NA; 27:10; Y; O; 1-week baseline + 8 weeks intervention	cocaine dependence (DSM-IV); NA; C	Brief Substance Craving Scale (BSCS); Cocaine Selective Severity Assessment (CSSA)	Oral varenicline 2 mg or placebo daily for 8 weeks; manual-guided cognitive behavioral therapy	BSCS: craving showed a significant decline over the trial in both groups (F = 20.34, *p* < 0.001); CSSA: both varenicline and placebo-treated groups show significant decreases in craving from baseline to end of study (*p* = 0.004)	NA; NA
Preston et al., 1993, USA [127]	18; 33; 18:0; N; I; 5 weeks	cocaine abusers/dependent (DSM-III); IV; C	VAS: rated ‘Desire for Cocaine’ by placing a mark along a 100-mm line marked at either end with none and extremely	Intravenous cocaine (0, 12.5, 25 and 50 mg) was administered 2 h after oral mazindol (0, 1 and 2 mg), 2 times per week for 5 weeks; NA	VAS: there were no significant main effects of mazindol alone and no significant mazindol/cocaine interactions on any scales on the VAS	55% (10); N
Price et al., 2012, USA [128]	32; 44; 22:10; Y,N; O; 3 days of intervention + 12 days observation	Cocaine dependence(DSM-IV); NA; C	VAS	Oral D-cycloserine 50 mg (DCS) only or D-cycloserine 50 mg +placebo (DSC-PBO) or placebo (PBO), for 3 sessions; NA	VAS: while baseline craving was not significantly different between groups at the beginning of session one, the session two baseline craving was significantly higher in the DCS-only group as compared to the PBO group and the session three baseline craving was significantly higher in the DCS-only group as compared to both the PBO and the DCS–PBO groups	NA; NA
Prisciandaro et al., 2013, USA [129]	25; 45; 23:2; NA; O; 2 weeks	Cocaine dependence (DSM-IV); A	Timeline Follow-back (TLFB): participants were asked to rate their craving, from zero (“none”) to four (“severe”)	Oral D-cycloserine (DCS) 50 mg or placebo for 2 sessions; two sessions of cocaine cue exposure and skills training	TLFB: there was no significant interaction between medication group and MRI session (*p* = 0.13; placebo participants’ craving from pre-scan to post-scan: Cohen’s d = −0.37; DCS participants’ craving from pre-scan to post-scan: Cohen’s d = 0.47).	18% (4); NA
Reid et al., 1998, USA [126]	20; NA; 18:2; NA; I; 2 days	Self-report patients with a history of smoking crack cocaine and cigarette smokers; S; C	VAS: a 1–100 mm scale demarcated craving at 25 mm intervals with the adjectives: not at all; mildly; moderately; and extremely	Single dose of transdermal nicotine 44 mg or placebo; NA	VAS: the cue-induced increase in cocaine craving was strongly enhanced by nicotine	NA; NA
Reid et al., 1999, USA [85]	23; 41; 20:3; NA; O; 2 days	Cocaine dependence (DSM-IV); S; C	VAS: cocaine craving and desire to use cocaine now	Single dose of oral mecamylamine 2.5 mg or placebo; NA	VAS: cue-induced cocaine craving and desire to use cocaine now were significantly lower during the mecamylamine condition	NA; NA
Reid, Angrist et al., 2005, USA [130]	23; 45; 18:5; NA; O; 8 weeks	Cocaine dependence (DSM-IV); S, I; C	Brief Substance Craving Scale (BSCS)	Oral celecoxib 200 mg (CLX) or placebo (PBO) daily for 8 weeks; individual cognitive behavioral counseling	BSCS: moderate decrease in craving when comparing baseline with last study visit. However, the change was not significantly different between celecoxib and placebo groups (Baseline CLX 6.3 ± 2.8, PBO 6.6 ± 2.6; End-point CLX 4.2 ± 2.3, PBO 5.4 ± 2.8 *p*= 0.686);	48% (11); NA
Reid et al., 2005, USA [131]	35; NA; NA; NA; O; 8 weeks of placebo + 16 week of active intervention	Methadone maintained subjects and cocaine dependence (DSM-IV); NA; C	Brief Substance Craving Scale (BSCS)	Transdermal mecamylamine 6 mg/ or placebo daily for 16 weeks; NA	BSCS: Cocaine craving showed moderate improvement in both groups, with a significantly greater reduction in cocaine craving (*p* < 0.05) in the placebo group	20% (7); Y
Renshaw et al., 1999, USA [132]	14; 37; 11:3; NA; O; 2 weeks	Cocaine dependence (DSM-IV); NA; C, A	VAS: How strong is your desire not to use cocaine right now? How strong is your urge for cocaine right now? CCQ (Cocaine Craving Questionnaire)	Oral citicoline 500 mg or placebo daily for 14 days; NA	CCQ: results revealed a group × condition interaction on the CCQ category, “Lack of control over use” (F1,8 = 6.02, *p* = 0.040), with citicoline treatment group reporting a decrease in “Lack of control over use” from pre- to post-treatments. There were no other significant differences in other measures; VAS: placebo group reported greater “Urge for cocaine” than the citicoline group at the post-treatment session, both prior to and following presentation of the crack cocaine cue video (F1,9 = 10.91, *p* = 0.01, and F1,9 = 16.62, *p* = 0.002, respectively). A main effect for condition “Desire to use cocaine right now” (F1,8 = 5.57, *p* = 0.046) was revealed, with subjects reporting a greater desire to use cocaine pre-treatment as compared to post-treatment, regardless of treatment or video presented.	NA; NA
Rosse et al., 1994, USA [133]	66; 33; 66:0; NA; I; 3 weeks	Cocaine dependence (DSM-III); NA; C	Questionnaires of cocaine craving and urges (QCU); VAS: rate craving from 0 (none) to 3 (severe)	Oral nimodipine (NDP) 90 mg or placebo (PBO) daily for 21 days; intensive 12-step milieu-oriented psychosocial therapy 4 h/day	QCU: significant decreases for all items, but no difference between groups (Baseline (Mean (SD): PBO: 64,8 (32,3), NDP: 60,4 (34,7) Endpoint: PBO: 44,8 (35,0), NDP: 47,3 (40,9)); VAS: craving was not significantly altered by the intervention (Baseline (Mean0–3): PBO: 0,62, NDP: 0,48; Endpoint: PBO: 0,50, NDP: 0,46)	NA; NA
Saladin et al., 2013, USA [134]	50; 40; 33:17; NA; I; 2 days of procedure + 1-week observation	Cocaine dependence (DSM-IV); S; C	Craving/Distress/Mood States Scale (CDMS): 100 mm visual analogue scales (VAS), with each being anchored by the adjectival modifiers “not at all” (left side of scale) and “extremely” (right side of the scale). The craving item asked participants to rate the desire to use cocaine “right now”	Single oral dose of propranolol 40 mg or placebo; NA	VAS After 24 h: propranolol treated group would evidence significantly lower cue-elicited cocaine craving and reactivity relative to the placebo treated group; after 1 week: craving was lower in the propranolol group, but this difference did not exceed threshold for significance/trend. Propranolol group evidenced significantly lower mean craving than the placebo group (F1,47 = 4.98, *p* = 0.03)	NA; NA
Santa Ana et al., 2015, USA [135]	47; 47; 40:7; NA; O; 1 week	Cocaine dependence (DSM- IV); NA; C	VAS: Craving was assessed by ‘do you crave cocaine? (from ‘0′ = “not at all” -‘10′ = “extremely”) of subjective desire to use cocaine	2 sessions of D-cycloserine 50 mg or placebo, with 1-week interval; NA	VAS: craving scores declined in both treatment conditions with differences becoming statistically significant from baseline	2% (1); NA
Schmitz et al., 2017, USA [136]	30; 48; 22:8; Y; O; 12 weeks	Cocaine dependence (DSM-IV); NA; NA	Brief Substance Craving Scale (BSCS); VAS: consisting of 100 mm line, anchored by 0 “not at all” and 100 “extremely,” for cocaine craving right now, craving on average in the past week, and the worst craving in the past week	Oral pioglitazone (PIO) 45 mg or placebo (PBO) daily for 12 weeks; weekly cognitive-behavioral therapy and prize-based contingency management for attendance	BSCS: reduced by a factor of 0.24 for participants receiving PIO compared to 0.09 for participants receiving PBO; VAS: decrease in craving by a factor of 3.84 for PIO versus 1.34 for PBO	40% (12); Y
Schmitz et al., 2001, USA [137]	85; 34; 62:23; NA; O; 12 weeks of intervention + 1-week observation	Cocaine dependence (DSM-IV); NA; NA	Cocaine Craving Scale (CCS)	Oral naltrexone 50 mg or placebo daily for 12 weeks; relapse prevention (RP) therapy or drug counseling	CCS: no differences on craving as a function of time, therapy, or medication. overall mean craving scores were lower at posttreatment (M= 16,1; SD = 24,3) than at intake (M = 29,7; SD = 32,3), consistent with a significant effect of time, F(2;37) = 3,57; *p* = 0,03	51% (43); Y
Schmitz et al., 2008, USA [138]	161; 41; 134:27; Y; O; 12 week	Cocaine dependence (DSM-IV); NA; C	Brief Substance Craving Scale (BSCS)	Oral sustained release levodopa/carbidopa 800/200 mg daily or placebo for 12 weeks; Clinical Management (ClinMan); ClinMan + Cognitive Behavioral Therapy (CBT); or ClinMan + CBT + Voucher-Based Reinforcement Therapy (VBRT)	BSCS: those receiving levodopa reported significantly lower craving scores (levodopa, M = 2.8, SE = 0.30; placebo, M = 3.7, SE = 0.31)	59% (95); Y
Schulte et al., 2018, Netherlands [139]	38; 18–55; 38:0; Y; O; 25 days	Cocaine use disorder (DSM-5); I; NA	Questionnaire for Cocaine Urges (QCU); Obsessive Compulsive Drug Use Scale (OCDUS); Desire for Drug Questionnaire (DDQ); VAS: ranging from 1 to 10 on which participants had to indicate their craving	Oral N-acetylcysteine (NAC) 2400 mg or placebo daily for 25 days; Working memory (WM) training online	Laboratory assessment: more positive effects of WM-sessions for the NAC group (B = 0.44 (0.10), *p*= 0.005) on craving, than for the placebo group (B= 0.04 (0.13), *p* =0.73). Ecological momentary assessment: no significant effects on craving of the treatment.	37% (14); N
Shoptaw et al., 2002, USA [140]	69; 36; 55:14; Y; O; 2 weeks screening +16 weeks intervention + brief follow up after 9 months of intervention	Cocaine dependence (DSM-IV); S, I, O; NA	VAS: (0‘not at all’ to 100‘strongest ever’) that asked participants to mark on a 100 mm line indicating their ‘‘most intense craving for cocaine that occurred at any time during the past 24 h’’	Oral amantadine hydrochloride 200 mg or placebo daily for 16 weeks; three times weekly, 90-min Matrix Model cocaine counseling sessions	VAS: placebo group reported craving levels at baseline that were almost two times greater than in amantadine group. This proportionate difference between conditions was maintained to treatment termination with a statistically significant effect of medication condition on craving ratings (F(1,64) = 7.27, *p* < 0.001) after including baseline ratings as covariates	78% (15); NA
Shoptaw et al., 2003, USA [141]	70; 35; 48:22; Y; O; 16 weeks	Cocaine dependence (DSM-IV); S, I, IV, O; C	VAS: “most intense craving for cocaine that occurred at any time during the past 24 h”	Oral baclofen 60 mg or placebo daily for 16 weeks; thrice-weekly cognitive-behavioral drug counseling groups	VAS: no statistically significant difference between participants	76% (53); NA
Winhusen et al., 2005, USA [100]	15; 40; NA; N; I; 2 days intervention+ 2 days observation	Cocaine dependence (DSM-IV); S, IV; C	VAS: 0–100 of desire for drug	Oral metyrapone 750 mg or placebo and infusion (40 mg of cocaine or saline) for 2 days; NA	VAS: the stress and conditioned craving procedures did not significantly alter participants’ VAS ratings; thus, the potential effects of metyrapone on stress- and cue-induced craving could not be evaluated	20% (3); NA
Winhusen, Somoza, Sarid-Segal et al., 2007, USA [142]	119; 41; 84:35; Y; O; 12 weeks	Cocaine dependence (DSM-IV); S, I; C	Brief Substance Craving Scale (BSCS)	Oral reserpine 0.5 mg or placebo daily for 12 weeks; weekly individual cognitive behavioral therapy	BSCS: both groups reported significantly less craving over the course of the study, but there was no significant difference between the groups as indicated by the non-significant Medication by Week (Z = −1.15, *p* > 0.05) and Medication (Z = −0.32, *p* > 0.05) effects	34% (40); NA
Woodcock et al., 2021, USA [143]	12; 48; 11:1; N; I; 2 weeks	Cocaine use (self report); S; C	Cocaine Craving Questionnaire (CCQ)	N-acetylcysteine 3600 mg or placebo daily for 1 week and then crossed over for 1 more week; NA	CCQ: no impact in craving	0; N

S1: study 1; S2: study 2.

## Data Availability

All the Appendix A are presented in the manuscript.

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
