# Peer review of "Pharmacological Treatments for Cocaine Craving: What Is the Way Forward? A Systematic Review"

_brainsci, 2022, doi:10.3390/brainsci12111546_

Round 1

Reviewer 1 Report

The manuscript makes a review that given the situation in the clinical field of cocaine craving treatment is of interest. However, the authors should do an in-depth review of the current version, taking into account my main concerns:

- The search strategy without temporal limitation is not very appropriate, since this has included papers with old drugs that are no longer marketed today and that all the world guidelines have established their non-efficacy. It would be appropriate to limit the time period from 2010, since between 2000 and 2015 there are reviews in this regard, albeit partial or focused on a type of drug (psychostimulants, antipsychotics, antidepressants). And it would be appropriate for the authors to add the last two years, so that the review is as up-to-date as possible.

- Related to the previous point, an established point of eligibility is the diagnosis with DSM-IV OR DSM-5, but in the old selected works there are many with DSM-III or DSM-III-R criteria (i.e. Stine et l., 1995; Crosby et al., 1996; Halikas et al., 1997; Buydens-Branchey et al., 1997; Chadwick et al. 1990; Eiler et al., 1995; Handelsman et al., 1995, 1997; Reid et al., 1998, 1999; Rosse et al., 1994; …). In the same way, the tables should be reviewed and DSM-5 and not DSM-V indicated, since it has been agreed to stop using Roman numerals. In relation to the diagnosis, if it has not been made with DSM-IV or DSM-5, as is the case in some studies, for consistency with the eligibility criteria it should not be included in the review either.

 The authors must also justify why they have not carried out a meta-analysis, which, based on the number of resulting works and their characteristics, seems feasible.

- Finally, it is necessary to carry out a careful review of the text, currently with different fonts, and to be consistent with the format of the references, since the surnames of authors and the year of publication are used in the text and they are numbered in the list.

Author Response

REVIEWER 1

The manuscript makes a review that given the situation in the clinical field of cocaine craving treatment is of interest. However, the authors should do an in-depth review of the current version, taking into account my main concerns:

Comment 1:

The search strategy without temporal limitation is not very appropriate, since this has included papers with old drugs that are no longer marketed today and that all the world guidelines have established their non-efficacy. It would be appropriate to limit the time period from 2010, since between 2000 and 2015 there are reviews in this regard, albeit partial or focused on a type of drug (psychostimulants, antipsychotics, antidepressants). And it would be appropriate for the authors to add the last two years, so that the review is as up-to-date as possible.

Response 1:

We have updated our research in the new manuscript, as pointed in this comment.

We actually prefer not to limit the time period because the studies in this field are very limited. We wanted to make a robust review of the subject.

Comment 2:

Related to the previous point, an established point of eligibility is the diagnosis with DSM-IV OR DSM-5, but in the old selected works there are many with DSM-III or DSM-III-R criteria (i.e. Stine et l., 1995; Crosby et al., 1996; Halikas et al., 1997; Buydens-Branchey et al., 1997; Chadwick et al. 1990; Eiler et al., 1995; Handelsman et al., 1995, 1997; Reid et al., 1998, 1999; Rosse et al., 1994; …). In the same way, the tables should be reviewed and DSM-5 and not DSM-V indicated, since it has been agreed to stop using Roman numerals. In relation to the diagnosis, if it has not been made with DSM-IV or DSM-5, as is the case in some studies, for consistency with the eligibility criteria it should not be included in the review either.

Response 2:

According to the previous response, we preferred to maintain the old studies to compose a more complete review. We have corrected the citations of DSM-5 that were in roman numerals.

Comment 3:

The authors must also justify why they have not carried out a meta-analysis, which, based on the number of resulting works and their characteristics, seems feasible.

Response 3:

This review is limited by the heterogeneity of craving assessment used across different studies, thereby preventing an accurate meta-analysis. Further, most studies had retention in treatment and abstinence as the main outcomes – whereas craving was considered a secondary outcome, with a lot of studies not describing exactly the method of evaluating the result of the craving measures.

Comment 4:

Finally, it is necessary to carry out a careful review of the text, currently with different fonts, and to be consistent with the format of the references, since the surnames of authors and the year of publication are used in the text and they are numbered in the list.

Response 4:

We have corrected the fonts and the format of the references.

Reviewer 2 Report

This is a a timely and well-done systemic review on an important topic. I applaud the authors and the team for examining an important clinical question. 

Comments:

I would suggest that the quality assessment and risk of bias table (Table 1), and Table 2 be moved to the supplement.

The search was conducted in August 2020. Search could be updated to include the last two years of published literature. (The  Cochrane Collaboration policy is that reviews should either be updated within two years or include a commentary to explain why this is not the case.)

Please include a completed/filled PRISMA checklist for the reviewers. The checklist for systematic reviews is available at  prisma-statement.org/PRISMAStatement/Checklist.aspx

Author Response

REVIEWER 2

This is a a timely and well-done systemic review on an important topic. I applaud the authors and the team for examining an important clinical question.

Comment 1:

I would suggest that the quality assessment and risk of bias table (Table 1), and Table 2 be moved to the supplement.

Response 1:

We have followed these suggestions.

Comment 2:

The search was conducted in August 2020. Search could be updated to include the last two years of published literature. (The  Cochrane Collaboration policy is that reviews should either be updated within two years or include a commentary to explain why this is not the case.)

Response 2:

We have updated our research in the new manuscript, as pointed by the 2 reviewers.

Comment 3:

Please include a completed/filled PRISMA checklist for the reviewers. The checklist for systematic reviews is available at  prisma-statement.org/PRISMAStatement/Checklist.aspx

Response 3:

We have attached the checklist.

Reviewer 3 Report

This is a comprehensive review of the pharmacological treatments of cocaine craving. The manuscript is well written considering the complexity of the cocaine use disorder. I have some suggestions for the authors:

1.     The authors analyzed not just cocaine dependent but also concurrent opiate and cocaine dependent individuals. I suggest to add this to study limitations.

2.     I am missing the information of buprenorphine on efficiency in the manuscript.  Regarding the combined pharmacotherapy Wee et al., 2012 suggest that the combination of buprenorphine and naltrexone at an appropriate dosage decreases compulsive cocaine self-administration with minimal liability to produce opioid dependence and may be useful as a treatment for cocaine addiction (https://doi.org/10.1126/scitranslmed.3003948). Several reports also indicated that buprenorphine decreases cocaine craving and use in opioid-dependent humans who were concurrently abusing cocaine (Schottenfeld et al., 1997; Foltin and Fischman, 1996; Montoya et al., 2004).

3.     I think it’s worth to mention hcrt/ox pharmacological antagonists potential in the discussion chapter.

4.     Chapter Results: line 171-173 – Please, correct the sum is not 100%.

5.     The references are not according the MDPI style.

Author Response

This is a comprehensive review of the pharmacological treatments of cocaine craving. The manuscript is well written considering the complexity of the cocaine use disorder. I have some suggestions for the authors:

Comment:
The authors analyzed not just cocaine dependent but also concurrent opiate and cocaine dependent individuals. I suggest to add this to study limitations.

Response:
We have added this in the limitations section (lines 1577-1579), as follows:
“Further, the majority of the studies considered abstinence retention and as the main outcomes in cocaine treatment, whereas craving was a secondary outcome; also some of the studies evaluated patients with cocaine use disorder with comorbidity with opioid or alcohol use disorder, schizophrenia, bipolar disorder or attention deficit hyperactivity”

Comment:
I am missing the information of buprenorphine on efficiency in the manuscript.  Regarding the combined pharmacotherapy Wee et al., 2012 suggest that the combination of buprenorphine and naltrexone at an appropriate dosage decreases compulsive cocaine self-administration with minimal liability to produce opioid dependence and may be useful as a treatment for cocaine addiction (https://doi.org/10.1126/scitranslmed.3003948). Several reports also indicated that buprenorphine decreases cocaine craving and use in opioid-dependent humans who were concurrently abusing cocaine (Schottenfeld et al., 1997; Foltin and Fischman, 1996; Montoya et al., 2004).

I think it’s worth to mention hcrt/ox pharmacological antagonists potential in the discussion chapter.

Response:
We have included such information in the discussion section of the present manuscript (lines 1529-1534; 1540-1542), as follows:
“Several studies have been conducted to evaluate the efficacy of pharmacotherapies regarding dual substance disorders: some reports indicated that buprenorphine decreases cocaine use [153,154,155] in opioid-dependent humans who were concurrently abusing co-caine, one in particular [155] evaluated craving and demonstrated a significant reduction in its rate. Recent studies examined the therapeutic potential of orexin receptor antagonists in rodent models of cocaine use disorder, in reducing cocaine seeking behavior [156,157], pointing to future directions in studies in clinical settings.”
“A pre-clinical study [159] suggests that the combination of buprenorphine and naltrexone decreases compulsive cocaine self-administration with minimal liability to produce opioid dependence and may be useful as a treatment for cocaine addiction.”

Comment:
Chapter Results: line 171-173 – Please, correct the sum is not 100%.

Response:
We have corrected this in the present manuscript.

Comment:
The references are not according the MDPI style.

Response:
We have corrected this in the present manuscript.

Round 2

Reviewer 1 Report

The authors have considered the least relevant of my comments related to the format of references.

In the current version there is still the problem of not limiting the time period, which, as I said, is based on the fact that the old papers have now been superseded and that dedicated to certain substances there are very adequate previous reviews, as the authors themselves detail in their text: Given the clinical relevance of craving as a potential mediator of return to using cocaine, there is growing interest in studying stimulants, antidepressants, and anticonvulsants as anti-craving pharmacotherapies, as reviewed in recent metaanalysis (Castells et al, 2007; Indave, Minozzi, Pani & Amato, 2016; Minozzi et al, 2015; Pani, Trogu, Vecchi & Amato, 2011).

 Note that there are two references from Castells et al. (2007) that in the text are not distinguished (a and b?)

 That is, what is the difference or concrete contribution of this revision? In my opinion, consider all the substances and an update regarding the previous works from the date I suggested. Although there is no page limitation in the Journal, the length of this manuscript is frankly unsustainable to be attractive to readers.

 – The justification for why a meta-analysis has not been carried out should not remain a mere response to my review, but included the justification also in the revised text, obviously. Although it is difficult not to have carried them out separately considering the type of substances if previous reviews have done so.

Author Response

We would like to thank the reviewer for the careful consideration of our manuscript. Please find below a point-by-point response to the comments. All the modifications are tracked in the revised manuscript. We are happy to make further adjustments if needed.

Comment 1:

The authors have considered the least relevant of my comments related to the format of references.

Response 1:

We have changed the format in this new version.

Comment 2:

In the current version there is still the problem of not limiting the time period, which, as I said, is based on the fact that the old papers have now been superseded and that dedicated to certain substances there are very adequate previous reviews, as the authors themselves detail in their text: Given the clinical relevance of craving as a potential mediator of return to using cocaine, there is growing interest in studying stimulants, antidepressants, and anticonvulsants as anti-craving pharmacotherapies, as reviewed in recent metaanalysis (Castells et al, 2007; Indave, Minozzi, Pani & Amato, 2016; Minozzi et al, 2015; Pani, Trogu, Vecchi & Amato, 2011).

Response 2:

The cited metaanalysis aimed to evaluate pharmacotherapies for cocaine use disorder, not specifically the treatment for the craving. Some of the studies included craving as an outcome, but not all of them, which was pointed out as an area of interest for future studies.

We have sustained the unlimited period of time, because some of the pharmacotherapies (e.g., desipramine) was Only studied at an older period of time (1995-2005) and some was only studied once (e.g. disulfiram – 2000). Therefore, we still think that is important for our extensive review.

Comment 3:

Note that there are two references from Castells et al. (2007) that in the text are not distinguished (a and b?) That is, what is the difference or concrete contribution of this revision? In my opinion, consider all the substances and an update regarding the previous works from the date I suggested.

Response 3:

The references were corrected. Until data, we do not have any review evaluating the anti-craving effect of these pharmacotherapies. We do have great reviews and metananalyses evaluating abstinence or cocaine use, but none of them evaluated systematically craving.

Comment 4:

Although there is no page limitation in the Journal, the length of this manuscript is frankly unsustainable to be attractive to readers.

Response 4:

Due to the long tables of details of the studies the article took too many pages. It is totally feasible to switch the tables 1-5 to supplementary materials, if the reviewers agree.

Comment 5:

The justification for why a meta-analysis has not been carried out should not remain a mere response to my review, but included the justification also in the revised text, obviously. Although it is difficult not to have carried them out separately considering the type of substances if previous reviews have done so.

Response 5:

We will include this justification.